# Plan in Sandbox, Navigate in Open Worlds: Learning Physics-Grounded Abstracted Experience for Embodied Navigation

**Zhixuan Shen** [1 2 †] **Jiawei Du** [2] **Ziyu Guo** [1] **Han Luo** [1 3] **Lilan Peng** [1] **Joey Tianyi Zhou** [2] **Haonan Luo** [1] **Tianrui Li** [1]

## Abstract

Vision-Language Models (VLMs) have demonstrated exceptional general reasoning capabilities. However, their performance in embodied navigation remains hindered by a scarcity of aligned open-world vision and robot control data. Despite simulators providing a cost-effective alternative for data collection, the inherent reliance on photorealistic simulations often limits the transferability of learned policies. To this end, we propose *Sandbox-Abstracted Grounded Experience* (*SAGE*), a framework that enables agents to learn within a physics-grounded semantic abstraction rather than a photorealistic simulation, mimicking the human capacity for mental simulation where plans are rehearsed in simplified physics abstractions before execution. *SAGE* operates via three synergistic phases: (1) *Genesis*: constructing diverse, physics-constrained semantic environments to bootstrap experience; (2) *Evolution*: distilling experiences through Reinforcement Learning (RL), utilizing a novel asymmetric adaptive clipping mechanism to stabilize updates; (3) *Navigation*: bridging the abstract policy to open-world control. We demonstrate that *SAGE* significantly improves planner-assisted embodied navigation, achieving a 53.21% LLM-Match Success Rate on A-EQA (+9.7% over baseline), while showing encouraging transfer to physical indoor robot deployment. Project page is available at: https://frankzxshen.github.io/SAGE.

[†]Work done while working at A*STAR as a visiting scholar. [1]School of Computing and Artificial Intelligence, Southwest Jiaotong University, China [2]Centre for Frontier AI Research, Agency for Science, Technology and Research (A*STAR), Institute of High Performance Computing, Agency for Science, Technology and Research (A*STAR), Singapore [3]School of Computer Science, University of Leeds, UK. Correspondence to: Tianrui Li <trli@swjtu.edu.cn>.

*Proceedings of the 43rd International Conference on Machine Learning*, Seoul, South Korea. PMLR 306, 2026. Copyright 2026 by the author(s).

## 1. Introduction

Recent rapid advancements in Vision-Language Models (VLMs) (Achiam et al., 2023; Liu et al., 2023; Yang et al., 2025a) have empowered agent systems with unprecedented capabilities in open-world perception, semantic reasoning, and zero-shot generalization. Motivated by these strides, the research community has increasingly focused on developing general-purpose embodied navigation agents. Broadly, these agents adhere to two primary paradigms. The first is goal-based navigation (Chang et al., 2023; Ziliotto et al., 2025; Yin et al., 2025), which requires the embodied agent to navigate to specific target object instances defined by class labels, reference images, or natural language descriptions. The other is Q&A-based navigation (Das et al., 2018; Majumdar et al., 2024; Ren et al., 2024; Yang et al., 2025b), where the agent actively gathers visual information through exploration to answer scene-related questions.

However, fully unleashing the potential of VLMs within embodied environments remains fraught with challenges. Unlike passive vision-language tasks (Radford et al., 2021) that benefit from Internet-scale datasets, embodied navigation suffers from a severe scarcity of aligned real-world vision perception and control data (O'Neill et al., 2024). Although the research community has turned to Reinforcement Learning (RL) to facilitate policy adaptation (Zeng et al., 2024; Choi et al., 2024; Wang & Huang, 2025) from high-level vision perception data to low-level robot control data, existing RL paradigms often falter in the Sim2Real transfer problem (Wong et al., 2025; Qureshi et al., 2025). Specifically, the huge modality gap between the semantic reasoning space of VLMs and the continuous actuation space of robots often renders learned policies brittle. When deployed in noisy real-world environments, these agents frequently fail to ground high-level intent plans into robust robot control. In the absence of structured guidance or embodied priors, RL agents are plagued by severe sample inefficiency and the cold-start problem. This creates a dilemma: while VLMs possess rich high-level reasoning knowledge but lack continuous learning capabilities for low-level robot control, RL provides a learning mechanism yet lacks the efficiency to learn effectively from scratch.

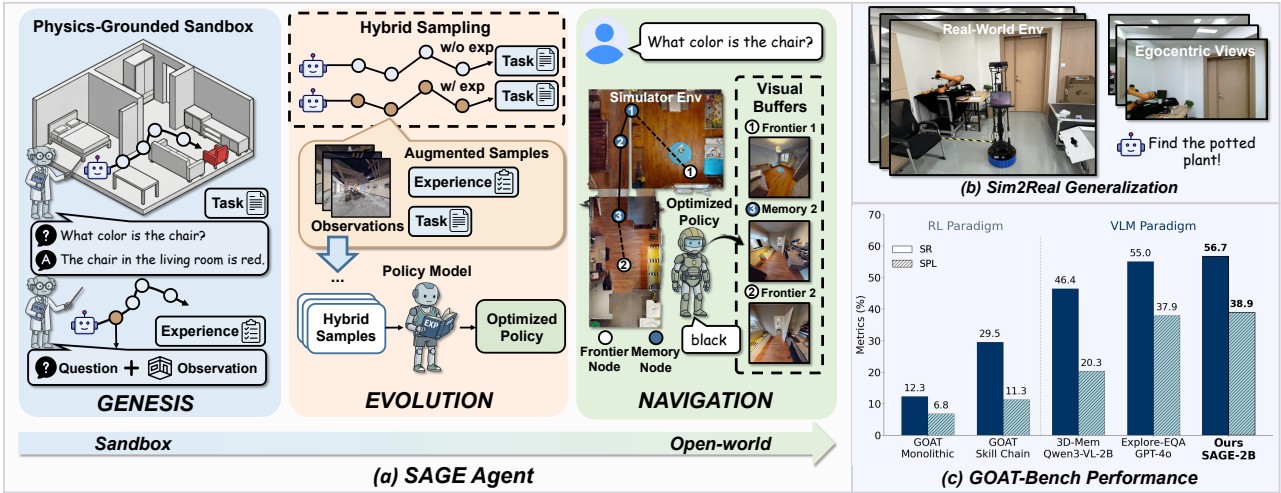

*Figure 1.* (a): Our *SAGE* framework utilizes a physics-grounded sandbox for self-evolving data generation and policy optimization, enabling the agent to bridge the gap between sandbox and open-world. (b): We demonstrate real-world robotic demonstrations powered by *SAGE*, showcasing its Sim2Real generalization capabilities. (c): Unlike other VLM or RL paradigms, our approach uniquely combines physics-grounded interactions with real-world transfer, demonstrating superior performance in navigation metrics.

Faced with this dilemma, our intuition is: ***since RL agents struggle to learn real-world policy from scratch, why not leverage the abstracted priors from the VLM perception in a sandbox environment?*** This process mirrors the human capacity for mental simulation before real-world execution. Instead of relying on difficult exploration in the real world, we propose operating the VLM within a physics-grounded sandbox to synthesize diverse tasks and proactively distill embodied experiences. Through this process, we can distill the VLM's high-level reasoning capabilities into the robot's policy in the form of massive, structured embodied experiences.

To resolve the challenges in the data-scarce embodied navigation, we propose a novel generative experience-driven learning paradigm named ***SAGE***. *SAGE* establishes a three-stage framework that enables agents to build robust navigation capabilities from scratch. Specifically, in *Genesis* phase, we mitigate data scarcity by autonomously synthesizing diverse tasks in a physics-grounded sandbox and distilling multi-view trajectories into embodied tasks and experiences. During *Evolution* phase, we introduce hybrid prompt-augmented sampling within the Group Relative Policy Optimization (GRPO) (Shao et al., 2024), imposing differentiated constraints on experience-augmented samples versus standard samples. Through an asymmetric adaptive clipping mechanism, the policy robustly internalizes retrieved priors while preserving training stability. Finally, the *Navigation* phase translates internalized semantic priors into planner-executable subgoals. Rather than learning end-to-end low-level motor commands, the learned policy selects task-relevant Frontier or Memory Nodes from structured visual buffers. These sub-goals are subsequently executed by

a geometric planner, effectively bridging the gap between semantic reasoning and robust low-level robot control.

Extensive experiments demonstrate that *SAGE* delivers strong performance across simulated embodied navigation benchmarks and provides encouraging evidence of transfer to physical indoor robot deployment. Through the proposed self-evolving framework, the agent continuously transforms synthetic sandbox priors into executable real-world policies, enabling sustained navigation performance enhancement. Consequently, *SAGE* achieves a 53.21% LLM-Match Success Rate on the A-EQA (Majumdar et al., 2024), surpassing the baseline approaches by a significant margin of 9.7%. Furthermore, we validate the practical efficacy of our approach through successful deployment on a physical robot, demonstrating robust transfer capabilities in real-world environments. In summary, the key contributions of our work are:

- We introduce a novel Generative Experience-Driven Learning paradigm to address the severe data scarcity and real-world transfer challenges in embodied navigation.

- We propose the *Genesis* phase that autonomously synthesizes diverse tasks in a physics-grounded sandbox and proactively leverages high-level VLM reasoning into structured embodied experiences.

- We develop the *Evolution* phase, featuring a hybrid prompt-augmented sampling strategy and an asymmetric adaptive clipping mechanism, which enables the policy to robustly internalize embodied priors while preserving training stability.

- We design the *Navigation* phase to bridge the intent-control gap. By decomposing high-level intents into Frontier or Memory Nodes, the robot effectively grounds abstract reasoning into robust navigation behaviors.

## 2. Problem Formulation

We formulate the learning problem via three core components: (1) a physics-grounded semantic sandbox $S$, which provides an environment $\mathcal{E}_S$; (2) an unknown target task distribution $\mathcal{O}$, representing diverse instructions autonomously synthesized by the agent; and (3) the downstream navigation task $\mathcal{N}$, denoting the capabilities the robot must master for real-world deployment. Our objective is to bridge the gap between the unsupervised sandbox $S$ and the high-level navigation task $\mathcal{N}$ by maximizing a surrogate objective over $\mathcal{O}$. We now define each component in detail.

### 2.1. Physics-Grounded Interaction Sandbox

**We formulate the interactive sandbox environment as a physics-grounded, reward-free Markov Decision Process (MDP).** Formally, it is represented as a tuple:

$$\mathcal{E}_S = (\mathcal{S}, \mathcal{A}, \mathcal{P}), \tag{1}$$

where $\mathcal{S}$ denotes the continuous state space encompassing 3D geometric and semantic information. $\mathcal{A}$ represents the agent's action space, which we decompose into the selection of discrete intermediate observations and their corresponding navigable waypoints. $\mathcal{P}(s'|s,a)$ denotes the state transition dynamics.

### 2.2. Target Task Distribution

We conceptualize $\mathcal{O}$ as a generative distribution of language-guided sandbox navigation tasks, approximating the diversity of real-world demands through autonomous synthesis. Formally, we define a task generation mapping $\phi : \mathcal{E}_S \rightarrow \mathcal{O}$, where each sampled task $o \in \mathcal{O}$ is a tuple $o = (I, \tau^*, a^*, \mathcal{K})$. Here, $I$ denotes a natural language query generated by the VLM; $\tau^*$ denotes a verified navigable trajectory; $a^*$ represents the ground-truth intermediate observation or answer for $I$; and $\mathcal{K}$ encapsulates extracted valid navigation knowledge (i.e., rules derived from successful exploration). As the robot explores the sandbox environment, increasingly complex tasks $o$ are populated into $\mathcal{O}$, thereby maximizing the diversity and complexity of the distribution.

### 2.3. Navigation Task

We formulate the navigation task $\mathcal{N}$ as a Goal-Conditioned Reinforcement Learning (GCRL) (Liu et al., 2022) problem. For any specific task $n \sim \mathcal{N}$, the agent aims to reach the target state via a policy $\pi_\theta(a|s,n)$, which maximizes the expected cumulative reward over the target distribution:

$$J_{\mathcal{N}}(\theta) = \mathbb{E}_{\substack{n \sim \mathcal{N}, \\ a_t \sim \pi_\theta(\cdot|s_t,n), \\ s_{t+1} \sim \mathcal{P}(\cdot|s_t,a_t)}} \left[ \sum_{t=0} \gamma^t r_{\mathcal{N}}(s_t, a_t, n) \right], \quad (2)$$

where $\gamma$ is the discount factor and $r_{\mathcal{N}}$ is the sparse ground-truth reward in the real world. Since $\mathcal{N}$ is unknown a priori and real-world sampling is prohibitively expensive, directly optimizing Eq. 2 is intractable. To address this, we propose approximating the optimal behavior by maximizing a surrogate objective $J_\phi(\theta)$ within the sandbox task distribution $\mathcal{O}$. Intuitively, the core objective is to optimize the policy against the synthesized experiences:

$$J_\phi(\theta) = \mathbb{E}_{\substack{o \sim \mathcal{O}, \\ a_t \sim \pi_\theta(\cdot|s_t,o), \\ s_{t+1} \sim \mathcal{P}(\cdot|s_t,a_t)}} \left[ \sum_{t=0} \gamma^t r_\phi(s_t, a_t, o) \right], \quad (3)$$

where $r_\phi$ is a shaped reward function designed to align sandbox behaviors with real-world requirements. By solving the surrogate optimization problem in Eq. 3 within the abstracted sandbox, the agent progressively acquires robust priors, ultimately allowing transfer to minimize the real-world objective $J_{\mathcal{N}}(\theta)$.

## 3. The Proposed *SAGE*

*SAGE* comprises three phases: (1) Genesis; (2) Evolution; (3) Navigation. We will sequentially introduce the core mechanisms and techniques of each phase in Sections 3.1, 3.2, and 3.3.

### 3.1. Genesis

In this section, we will detail the generation of Sandbox Tasks and Experience Rules, as shown in Figure 2 (a).

**Sandbox Environment Setup.** To ensure broad generalization, we instantiate the sandbox environment $\mathcal{E}_S$ using large-scale indoor datasets HM3D (Yadav et al., 2023) and InteriorGS (SpatialVerse Research Team, 2025). While these environments provide high-fidelity RGB-D data, directly learning from raw pixels is inefficient. Instead, we treat the environment as a graph of semantic states. We parse the continuous space into discrete navigable nodes, where the state transition dynamics $\mathcal{P}$ strictly obey physical constraints (e.g., collision-free traversability), ensuring that the synthesized plans are physically executable.

**Task Synthesis.** We establish an automated pipeline to populate the task distribution $\mathcal{O}$. First, we randomly sample start-goal pairs within navigable regions. For each pair, the A* algorithm computes the optimal geodesic trajectory $\tau^*$. To simulate the "Visual Options" action space $\mathcal{A}$, we discretize $\tau^*$ into $T$ keypoints. At each keypoint $t$, we render a set of intermediate observations

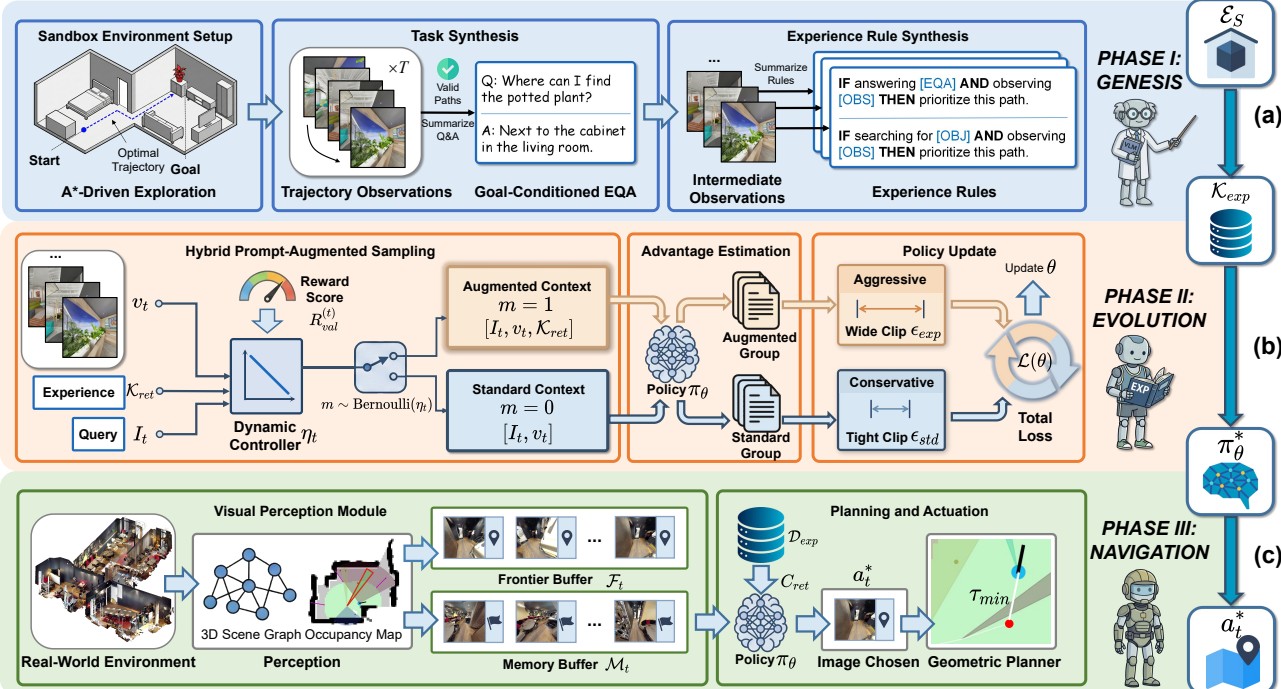

*Figure 2.* **The SAGE Framework.** The system operates in three phases: (a) **Genesis:** A sandbox environment $\mathcal{E}_S$ synthesizes task-oriented experience rules $\mathcal{K}_{exp}$. (b) **Evolution:** The policy $\pi_\theta$ is optimized via a hybrid prompt-augmented sampling strategy, utilizing both standard and augmented contexts. (c) **Navigation:** The embodied navigation policy relies on evolving policy $\pi_\theta$ and retrieved experience $C_{ret}$ to execute $a_t^*$ in complex indoor environments.

$\mathcal{V}_t = \{v_{t,0°}, v_{t,+120°}, v_{t,-120°}\}$, representing the forward view and potential branching frontiers (left/right).

$$\tau^*(\text{OBS}) = \{(v_{t,0°}, v_{t,+120°}, v_{t,-120°})\}_{t=1}^T, \quad (4)$$

the forward view $v_{t,0°}$ serves as the ground-truth action $a_t^*$ for the current step. Subsequently, we project detected objects into a semantic scene graph. Based on the trajectory endpoint and visible entities, a VLM synthesizes a natural language instruction $I$ and the corresponding answer. This process autonomously constructs the task tuple $o$. Detailed descriptions are provided in Appendix E.3.

**Experience Rule Synthesis.** To explicitly extract navigation knowledge $\mathcal{K}$, we analyze the causal relationship between visual observations and expert decisions within $\tau^*$. We formulate this as an *IF-THEN* reasoning rule: "*IF answering/searching [Task] AND observing [Scene Description] THEN prioritize this path.*". Specifically, for each step in $\tau^*$, we prompt the VLM to articulate the rationale for choosing $v_{t,0°}$ over other frontiers. These structured rules are encoded and stored in a vector database $\mathcal{D}_{exp}$, constructing a queryable knowledge base that supports the robot's learning paradigm in the subsequent *Evolution* phase.

### 3.2. Evolution

In this section, we elaborate on the optimization policy of *Evolution* phase, as shown in Figure 2 (b).

**Reward Shaping.** We define the reward as a weighted sum of format compliance and semantic alignment:

$$r_\phi(s_t, a_t) = w_f \mathbb{I}_f$$
$$+ w_{acc} \left( \mathbb{I}_m (1 + \text{sim}(a_t, a_t^*)) - \mathcal{P}_{err} \right), \quad (5)$$

where $\mathbb{I}_f$ is the indicator function for template compliance, and $\mathbb{I}_m$ indicates correct image selection logic. The term $\text{sim}(a_t, a_t^*)$ measures the textual similarity between the agent's output $a_t$ and the ground-truth output $a_t^*$. Finally, $\mathcal{P}_{err}$ imposes penalties for classification errors or formatting failures.

**Hybrid Prompt-Augmented Sampling.** To balance the trade-off between exploiting known successful trajectory experiences and exploring novel solutions, we employ a dynamic controller that injects retrieved experience $\mathcal{K}_{ret} \in \mathcal{K}_{exp}$ into the context with a time-varying ratio $\eta_t$. To ensure that $\eta_t$ is always a valid Bernoulli probability, we compute:

$$\eta_t = \max\left(\eta_{min}, \eta_{init} \cdot \left(1 - \frac{\min(R_{val}^{(t)}, R_{target})}{R_{target}}\right)\right), \quad (6)$$

where $R_{val}^{(t)}$ denotes the best reward score observed on the validation set, updated every 5 training steps, and $R_{target}$ is the target validation reward used by the schedule. At each interaction step, we sample a Bernoulli mask $m \sim$

Bernoulli($\eta_t$). The input context $x_t$ is constructed as:

$$x_t = \begin{cases} [I_t, v_t, \mathcal{K}_{ret}] & \text{if } m = 1 \\ [I_t, v_t] & \text{if } m = 0 \end{cases}, \qquad (7)$$

where $v_t \in \mathcal{V}$ represents a subset of length 4 from the RGB observation sequences $\mathcal{V}$ containing the ground-truth image, accompanied by object detection information. $m = 1$ corresponds to an augmented sample and $m = 0$ to a standard sample.

**Homogeneous Group Advantage Estimation.** To ensure unbiased credit assignment, we adopt a group-based relative policy optimization framework. Crucially, to prevent the naturally higher rewards of augmented samples from skewing the baseline for standard samples, we implement homogeneous grouping. For a given input $x_i$, we sample a group of $G$ outputs $\{a_{(i,1)}, \ldots, a_{(i,G)}\}$ and ensure that the mask $m_i$ is consistent within the group. The advantage $A_{i,j}$ for the $j$-th output in group $i$ is standardized strictly within its own distribution:

$$A_{i,j} = \frac{r_\phi(x_i, a_{i,j}) - \mu(\{r_{\phi_{i,k}}\}_{k=1}^G)}{\sigma(\{r_{\phi_{i,k}}\}_{k=1}^G) + \epsilon}, \qquad (8)$$

where $\mu$ and $\sigma$ represent the mean and standard deviation of rewards within the homogeneous group, $\epsilon$ is a small constant introduced for numerical stability.

**Policy Update.** The core of our optimization lies in differentiating the constraint strictness for knowledge absorption versus stability. Specifically, we encourage the policy to adapt aggressively when learning from augmented samples that exhibit high-reward behaviors. In contrast, updates derived from standard samples are deliberately kept conservative to reduce the risk of unstable training dynamics or policy collapse. However, high-quality samples might be excessively penalized due to reward noise or group normalization artifacts. To mitigate this issue, we propose an *Asymmetric Adaptive Clipping (AAC)* strategy. We denote the importance sampling ratio as $\rho_{i,t}(\theta) = \pi_\theta(a_{i,t}|x_{i,t})/\pi_{\theta_{old}}(a_{i,t}|x_{i,t})$. The dynamic upper clipping threshold $\epsilon_{up}(m_i)$ is defined based on the sample source:

$$\epsilon_{up}(m_i) = \begin{cases} \epsilon_{exp} & \text{if } m_i = 1 \\ \epsilon_{std} & \text{if } m_i = 0 \end{cases}, \qquad (9)$$

where $\epsilon_{exp} \gg \epsilon_{std}$. Unlike GRPO/PPO with symmetric boundaries, our objective anchors the lower bound to the conservative $\epsilon_{std}$ for all samples, while relaxing the upper

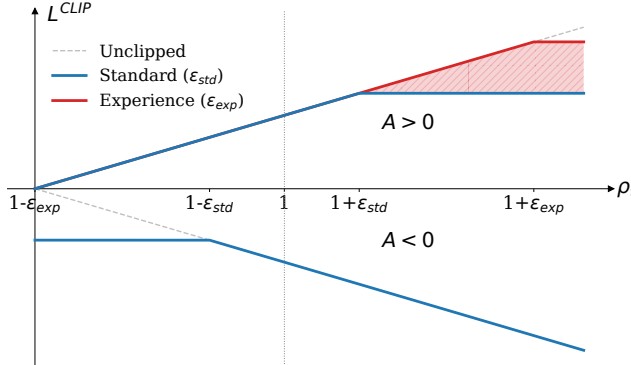

*Figure 3.* **Asymmetric Adaptive Clipping (AAC).** While both standard and augmented samples share a conservative lower bound $(1 - \epsilon_{std})$ to prevent policy collapse under $A < 0$, augmented experience samples feature an expanded upper bound $\epsilon_{exp}$ under $A > 0$. The shaded region indicates the additional optimization space allocated for aggressive knowledge absorption from high-reward augmented trajectories.

bound dynamically:

$$L_{i,t}^{CLIP} = \min \Big( \rho_{i,t} A_{i,t},$$
$$\text{clip}(\rho_{i,t}, 1 - \epsilon_{std}, 1 + \epsilon_{up}(m_i)) A_{i,t} \Big), \quad (10)$$
$$J_\phi(\theta) = \mathbb{E}_{(x,a) \sim \mathcal{B}} \Big[ L^{CLIP}(\theta)$$
$$- \beta \, \mathbb{D}_{KL}(\pi_\theta(x) \,\|\, \pi_{\theta_{ref}}(x)) \Big], \qquad (11)$$

where $L^{CLIP}$ is the clipped surrogate objective, $\pi_{\theta_{ref}}$ is the reference model, and the expectation is taken over transitions sampled from the current rollout batch $\mathcal{B}$. We provide the detailed theoretical analysis in Appendix F. The KL divergence term prevents excessive deviation from the foundation model. By employing this design, we enable the policy to use a more lenient $\epsilon_{exp}$ for substantial gradient updates to rapidly internalize experience. Conversely, by enforcing the tight lower bound $1 - \epsilon_{std}$ universally, we ensure that even if a "golden" augmented sample receives a negative advantage estimate due to variance, the policy remains protected against drastic probability collapse.

### 3.3. Navigation

As shown in Figure 2 (c), during the *Navigation* phase, we employ a retrieval-augmented embodied navigation paradigm for exploration and decision-making.

**Perception.** The navigation cycle initiates with the visual perception module, which transforms raw sensory data into structured semantic representations. At time step $t$, the robot receives RGB-D observations from the environment. We maintain a dynamic 3D scene graph and an occupancy map to incrementally aggregate environmental geometric information. To facilitate VLM reasoning, we abstract con-

tinuous data into two discrete visual buffers: the Memory Buffer $\mathcal{M}_t$ and the Frontier Buffer $\mathcal{F}_t$. $\mathcal{M}_t$ stores memory images and nodes of unique objects and landmarks observed along the trajectory, providing historical context for decision-making regarding question answering or target identification. Simultaneously, we compute the boundaries between free space and unexplored space to generate $\mathcal{F}_t$, which provides a set of candidate frontier images and nodes for the robot's exploration operations. Formally, the robot state representation $\mathcal{R}_t$ at each time step is constructed as:

$$\mathcal{R}_t = \{I_{query}, \mathcal{M}_t, \mathcal{F}_t\}, \qquad (12)$$

where $I_{query}$ denotes the instruction or image goal.

**Experience Augmentation.** To effectively leverage sandbox experience, we extract the most relevant experience $C_{ret}$ from the vector database $\mathcal{D}_{exp}$ and synthesize a composite prompt based on current observations and retrieved knowledge. The VLM policy $\pi_\theta$ aims to maximize the probability of the optimal action:

$$a_t^* = \underset{a \in \mathcal{A}}{\arg\max} \, \pi_\theta(a \mid \mathcal{R}_t, C_{ret}) \qquad (13)$$

where $\mathcal{A} = \mathcal{F}_t \cup \mathcal{M}_t$.

**Planning and Actuation.** Once a target from the buffers is selected, it is translated into low-level robot control commands via a geometric planner. For example, if the VLM agent selects a frontier image/node $F_i \in \mathcal{F}_t$ for exploration, the robot navigates from its current pose $x_t$ to $F_i$. In simulation, we employ the follower provided by Habitat-Sim (Szot et al., 2021) to track the shortest path $\tau_{min}$; in real-world deployment, we seamlessly switch to the ROS navigation stack utilizing a top-down occupancy map. To simulate real robot dynamics and maintain perceptual updates, the maximum step size during movement is limited to $\delta_{max} = 1.0$m. The movement for the current step terminates when the agent travels a distance of $\delta_{max}$ or reaches within a proximity threshold 0.5m of the target node $F_i$. This execution loop continuously updates $\mathcal{R}_t$ and queries the VLM policy until the agent selects an image $M_i \in \mathcal{M}_t$ as the final navigation goal or the maximum number of episode steps is reached.

## 4. Experiments

### 4.1. Experimental Settings

**Dataset and Evaluation Metrics.** We evaluate *SAGE* on two long-horizon embodied navigation benchmarks: **A-EQA** (Majumdar et al., 2024) and **GOAT-Bench** (Khanna et al., 2024). **A-EQA:** Designed for active exploration and question answering, the A-EQA dataset comprises 557 natural language queries across 63 real-world indoor scenes. Following the evaluation protocol established by 3D-Mem (Yang et al., 2025b), we report results on the

widely-used validation subset of 184 questions. Adhering to the OpenEQA (Majumdar et al., 2024) standards, we quantify performance using LLM-Match Success Rate (SR[†]) and LLM-Match Success weighted by Path Length (SPL[†]), utilizing `Qwen3-235B-A22B` (Yang et al., 2025a) as the automated evaluator. For SR[†], let $\hat{a}_i$ be the predicted answer and $a_i^*$ be the ground truth for the $i$-th question. The scoring function $\mathcal{F}_{LLM}(\hat{a}_i, a_i^*)$ returns a raw score in the range $[1, 5]$. The normalized SR[†] is defined as:

$$\text{SR}^{\dagger} = \frac{1}{N} \sum_{i=1}^{N} \left( \frac{\mathcal{F}_{LLM}(\hat{a}_i, a_i^*) - 1}{4} \right). \qquad (14)$$

SPL[†] scales the normalized correctness score by the ratio of the shortest path $l_i$ to the agent's actual path $p_i$:

$$\text{SPL}^{\dagger} = \frac{1}{N} \sum_{i=1}^{N} \frac{\mathcal{F}_{LLM}(\hat{a}_i, a_i^*) - 1}{4} \cdot \frac{l_i}{\max(p_i, l_i)}. \qquad (15)$$

In cases of navigation failure, the agent defaults to blind guessing; the contribution to SPL[†] is set to 0. **GOAT-Bench:** This benchmark challenges robots to sequentially execute 5 to 10 subtasks within unseen real-world scenes. Subtasks involve diverse modalities, including category names, detailed descriptions, or object images. Following the evaluation protocol established by 3D-Mem, we conduct evaluations on a subset of the *Val Unseen* split in the main text, totaling 278 subtasks across 36 scenes. GOAT-Bench employs standard Success Rate (SR) and Success weighted by Path Length (SPL). A task is considered successful only if the agent's final position is within $1.0$ m of the target.

**Baselines.** We benchmark *SAGE* against a diverse set of state-of-the-art (SOTA) methods categorized into two paradigms: (1) RL Paradigm, including SenseAct-NN variants (Khanna et al., 2024); and (2) VLM Paradigm, covering both closed-source models (e.g., Explore-EQA (Ren et al., 2024) and 3D-Mem (Yang et al., 2025b) powered by GPT-4o (Hurst et al., 2024)) and open-source local models (e.g., LLaVA-7B (Liu et al., 2023)). Implementation details and full-set evaluation are provided in Appendix A and H. Furthermore, we demonstrate the system's practical robustness via Real-World Deployment in Appendix J. All results reported are averaged over 3 independent random seeds. Due to space constraints, we report mean values in the main text.

### 4.2. Main Navigation Results

Table 1 presents the comparative results across diverse paradigms. *SAGE* demonstrates superior performance, significantly outperforming traditional RL baselines by a large margin. When controlling for model capacity, our 2B model significantly outperforms the 3D-Mem baseline with the identical backbone, achieving gains of +8.9% on A-EQA and +10.3% on GOAT-Bench, while nearly doubling the

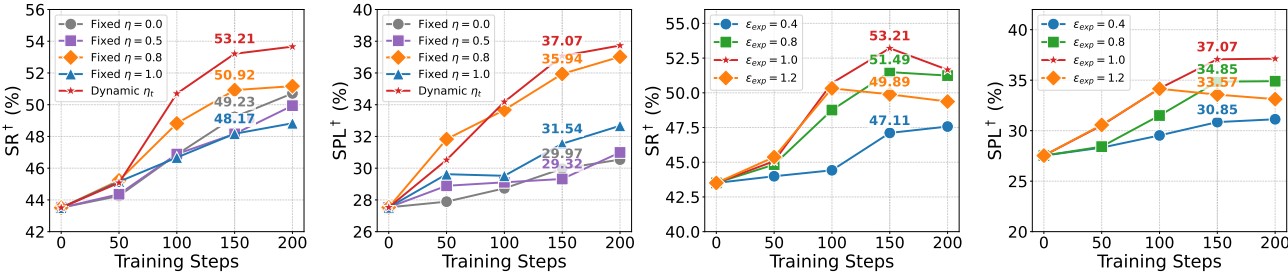

*Figure 4.* (a)&(b): Impact of fixed and dynamic experience-injection probabilities on navigation performance. We compare fixed $\eta \in \{0.0, 0.5, 0.8, 1.0\}$ with a validation-dependent dynamic schedule, where the red star curve uses $\eta_{\mathrm{init}} = 0.8$, $\eta_{\min} = 0.0$, and $R_{\mathrm{target}} = 1.5$. (c)&(d): Impact of upper clipping threshold $\epsilon_{exp}$ on navigation performance. All experiments use the model with 2B parameters on A-EQA.

*Table 1.* **Performance of *SAGE* on A-EQA and GOAT-Bench.** A-EQA results include both SR$^\dagger$ (Eq. 14) and SPL$^\dagger$ (Eq. 15). Methods with * are reported from OpenEQA (Majumdar et al., 2024) and 3D-Mem (Yang et al., 2025b). All values are in percent (%). **Bolded numbers** highlight the best open-source results.

| Method | A-EQA | | GOAT-Bench | |
|---|---|---|---|---|
| | SR$^\dagger$ | SPL$^\dagger$ | SR | SPL |
| ***RL Baselines*** | | | | |
| SenseAct-NN Skill Chain | 24.7 | 13.3 | 29.5 | 11.3 |
| SenseAct-NN Monolithic | 20.6 | 10.1 | 12.3 | 6.8 |
| ***Closed-Source VLMs*** | | | | |
| Explore-EQA* | 46.9 | 23.4 | 55.0 | 37.9 |
| 3D-Mem* (GPT-4o) | 52.6 | 42.0 | 69.1 | 48.9 |
| ***Open-Source VLMs*** | | | | |
| 3D-Mem (Qwen3-2B) | 44.3 | 19.4 | 46.4 | 20.3 |
| 3D-Mem* (LLaVA-7B) | - | - | 49.6 | 29.4 |
| *SAGE* (Qwen3-2B) (Ours) | 53.2 | 37.1 | 56.7 | 38.9 |
| *SAGE* (Qwen3-4B) | **60.2** | **47.2** | **64.8** | **44.9** |

SPL efficiency. Remarkably, despite its compact size, *SAGE* (2B) even surpasses the closed-source giant 3D-Mem (GPT-4o) in terms of A-EQA SR$^\dagger$, proving that internalizing sandbox-derived priors enables open-source models to rival proprietary leaders. Furthermore, scaling to 4B parameters, *SAGE* sets a new SOTA on A-EQA with a score of 60.2%. On the more complex GOAT-Bench, *SAGE* (4B) narrows the gap with GPT-4o to within 4.3%, validating the scalability of our learning paradigm.

### 4.3. Analysis on Sandbox Data

**Impact of Synthetic Data Composition.** We investigate how the diversity of the sandbox environment influences the efficacy of our *Evolution* phase. As illustrated in Figure 5 (a), the MIX strategy achieves the highest performance with an SR$^\dagger$ of 53.21%, surpassing the individual HM3D and InteriorGS. This result indicates that scene diversity and texture variety are critical for generalization. By integrat-

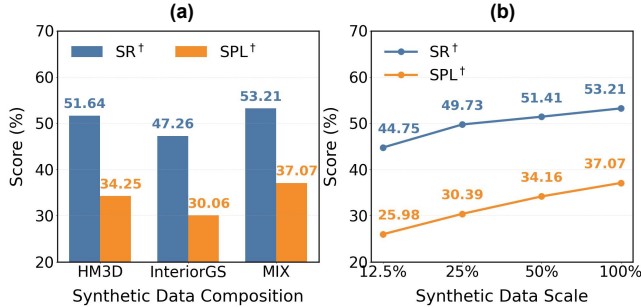

*Figure 5.* (a): Comparison of data composition strategies. (b): Impact of data scale on model performance. All experiments use the model with 2B parameters on A-EQA.

ing complementary environments during the *Genesis* phase, the agent learns more robust navigation priors that generalize better to unseen test scenarios, effectively preventing overfitting to specific simulator artifacts.

**Scalability with Minimal Data.** To assess the efficacy of sandbox data for policy evolution, we investigate the impact of synthetic data scale. Benchmarking against the set of 14,526 efficiently synthesized valid trajectories (comprising 7,988 from HM3D and 6,538 from InteriorGS) as the 100% baseline, Figure 5 (b) reveals a clear positive correlation between navigation performance and data volume. This empirically corroborates our core hypothesis in Section 3.1. Notably, we observe a trend of diminishing marginal returns; while the performance gain slows from 50% to 100% data scale, the trajectory efficiency SPL$^\dagger$ continues to improve (from 34.16% to 37.07%). Crucially, even when utilizing only 12.5% of the available data, the model achieves a respectable SR$^\dagger$ of 44.75%. This confirms that our approach effectively mitigates the dependency on massive datasets, paving a scalable avenue for enhancing navigation policies by integrating abundant, low-cost sandbox tasks.

### 4.4. Analysis on *Evolution*

**Model-dependent vs. Experience-dependent.** Heavy reliance on priors yields quick rewards but risks constrain-

*Table 2.* **Ablation on Visual Context Length.** We analyze the impact of the number of input frames $v_t$ on navigation performance. $v_t = 4$ yields the best balance between reasoning and efficiency.

| # Images ($v_t$) | SR$^\dagger$ (%) | SPL$^\dagger$ (%) |
|---|---|---|
| 2 | 46.29 | 29.18 |
| 3 | 49.72 | 33.82 |
| **4 (Ours)** | **53.21** | **37.07** |
| 5 | 53.45 | 36.67 |

ing long-term optimization. As shown in Figure 4 (a-b), we compare fixed experience-injection probabilities $\eta \in \{0.0, 0.5, 0.8, 1.0\}$ with our validation-dependent dynamic schedule, where $\eta_{\text{init}} = 0.8$, $\eta_{\text{min}} = 0.0$, and $R_{\text{target}} = 1.5$. Small fixed $\eta$ values provide insufficient prior guidance, leading to slower improvement, while overly large fixed $\eta$ makes the policy rely heavily on retrieved experiences and may constrain generalization. In contrast, the dynamic schedule preserves experience guidance in early training and gradually anneals $\eta_t$ as validation performance improves, achieving the best SR$^\dagger$ and SPL$^\dagger$. This confirms that dynamic experience injection guides the agent's transition from mimicking retrieved priors to autonomous exploration.

**Conservative vs. Aggressive Optimization.** Figure 4 (c-d) highlights the stability-plasticity trade-off controlled by $\epsilon_{exp}$. Conservative clipping ($\epsilon_{exp} = 0.4$) causes underfitting, failing to exploit *Genesis* signals. Conversely, aggressive updates ($\epsilon_{exp} = 1.2$) lead to instability and performance degradation after 100 steps. An optimally relaxed bound ($\epsilon_{exp} = 1.0$) balances this, enabling rapid experience absorption without policy collapse, achieving a peak SR$^\dagger$ of 53.21% and SPL$^\dagger$ of 37.07% at 150 training steps. This validates that a moderately relaxed upper bound is essential for bridging the gap between imitation and generalization.

**Sparse Observation vs. Rich Context.** To determine the optimal context window size for the VLM, we evaluated the agent's performance by varying the number of input frames $v_t$. As detailed in Table 2, increasing input frames $v_t$ from 2 to 4 significantly boosts performance, proving that sparse cues are insufficient. However, extending to 5 frames yields diminishing returns (SPL drops to 36.67%), suggesting that redundant visual tokens may dilute the VLM's attention without adding actionable information. Consequently, we adopt $v_t = 4$ as the default setting to balance reasoning capability with input efficiency.

### 4.5. Analysis and Ablation

**Analysis on Main Components.** As shown in Table 3, the full *SAGE* framework achieves substantial improvements of 9.70%/6.03% on A-EQA and 7.52%/8.09% on GOAT-Bench compared to the baselines (Qwen3-VL-2B/4B). Sim-

*Table 3.* **Effects of Main Components.** "Task" denotes synthesized tasks; "Exp" denotes experience rules; "AAC" denotes Asymmetric Adaptive Clipping. $C_{ret}$ indicates experience retrieved during navigation inference. All values are in percent (%).

| Method | A-EQA | | GOAT-Bench | |
|---|---|---|---|---|
| | SR$^\dagger$ | SPL$^\dagger$ | SR | SPL |
| ***Qwen3-VL-2B*** | 43.51 | 27.53 | 49.17 | 30.77 |
| $+C_{ret}$ | 46.47 | 30.72 | 50.58 | 30.87 |
| +Task | 50.71 | 33.68 | 53.72 | 34.64 |
| +Task+Exp | 51.42 | 34.67 | 54.05 | 38.52 |
| +Task+Exp+AAC | 51.88 | 36.29 | 55.35 | 38.59 |
| *SAGE* (Full) | **53.21** | **37.07** | **56.69** | **38.90** |
| ***Qwen3-VL-4B*** | 54.13 | 36.20 | 56.72 | 43.99 |
| $+C_{ret}$ | 55.87 | 40.03 | 57.04 | 44.41 |
| +Task | 57.14 | 45.70 | 61.19 | 42.90 |
| +Task+Exp | 58.42 | 45.74 | 63.50 | 43.57 |
| +Task+Exp+AAC | 59.03 | 46.67 | 64.28 | 44.85 |
| *SAGE* (Full) | **60.16** | **47.21** | **64.81** | **44.89** |

*Table 4.* **Ablation of *Navigation* Phase.** G.&E. represents the policy trained via *Genesis* and *Evolution* phases. $\mathcal{D}_{exp}$ is the experience database. Random Exp. denotes randomly selected experience. All values are in percent (%).

| Method | A-EQA | | GOAT-Bench | |
|---|---|---|---|---|
| | SR$^\dagger$ | SPL$^\dagger$ | SR | SPL |
| Zero-shot VLM | 43.51 | 27.53 | 49.17 | 30.77 |
| +G.&E. (w/o $\mathcal{D}_{exp}$) | 49.80 | 31.41 | 51.11 | 31.87 |
| +G.&E. (w/ Random Exp.) | 51.73 | 34.84 | 52.63 | 32.58 |
| *SAGE* (Full) | **53.21** | **37.07** | **56.69** | **38.90** |

ply injecting retrieved context during navigation ($+C_{ret}$) yields immediate zero-shot benefits (+1.74% on 4B), validating the quality of our sandbox knowledge. Subsequently, the introduction of sandbox-synthesized tasks (+Task) serves as a primary driver of capability, further raising A-EQA scores to 50.71% (2B) and 57.14% (4B). Building on this, the integration of Experience Rules coupled with our Asymmetric Adaptive Clipping strategy (+Exp&+AAC) yields consistent additional gains of 1.17% (2B) and 1.89% (4B) on A-EQA. This confirms that our optimization strategy successfully internalizes retrieved priors into the parametric policy. Ultimately, the full *SAGE* framework achieves peak performance by equipping this optimized policy with $C_{ret}$, demonstrating a synergistic effect where the policy learns *how* to reason while real-time retrieval provides *what* to reason about.

**Ablation for *Navigation*.** Table 4 validates the synergy between internalized policy and external retrieval in *Navigation* phase. Training via *Genesis* and *Evolution* (+G.&E.) boosts A-EQA SR$^\dagger$ by 6.29% over the zero-shot baseline,

confirming that the policy effectively internalizes sandbox priors even without inference-time retrieval. Furthermore, the full *SAGE* framework outperforms random experience injection. This demonstrates that while training establishes the reasoning capability, retrieving relevant experience is essential for maximizing success in complex environments.

## 5. Conclusion

We presented *SAGE*, an experience-driven learning paradigm designed for open-world generalization in embodied navigation. By leveraging the physics-grounded sandbox to synthesize tasks and abstract experiences, our framework effectively mitigates data scarcity. Through the synergistic phases of *Genesis*, *Evolution*, and *Navigation*, *SAGE* internalizes high-level intent into robust low-level control. Empirically, SAGE not only showcases its superiority in navigation benchmarks but also demonstrates exceptional generalization in open-world environments.

## Acknowledgment

This research was supported by the National Natural Science Foundation of China (No. U2468207, 62306247), Sichuan Science and Technology Program (No. 2024NSFTD0036), and the Fundamental Research Funds for the Central Universities under Grant 2682026ZT007. This research was supported by the Shanghai Science and Technology Program (Grant No. 25HB2703100), and the National Research Foundation, Singapore and Infocomm Media Development Authority under its Trust Tech Funding Initiative. Any opinions, findings, conclusions, or recommendations expressed in this material are those of the author(s) and do not reflect the views of the National Research Foundation, Singapore or Infocomm Media Development Authority.

## Impact Statement

Although our *SAGE* leads significant performance improvements, it does not guarantee perfect prediction. Therefore, significant attention must be paid for rigorous verification processes, prior to integrating it into the embodied AI system.

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

# A. Implementation Details

## A.1. Genesis

We employ `yolov8x-world` (Cheng et al., 2024b) as the object detection tool, synthesizing tasks and experiences based on `Qwen3-VL-Plus` (Yang et al., 2025a) with default settings and a sampling temperature of 1.0. For the sandbox environment, we set a uniform rendering resolution of $256 \times 256$, a Horizontal Field of View (HFOV) of $120°$, and a Sensor Height of 1.5m. For each trajectory, in addition to key points, we randomly sample 1 to 9 nodes by default to obtain intermediate observations.

## A.2. Evolution

During hybrid prompt-augmented sampling, we compare fixed experience-injection probabilities with a validation-dependent dynamic schedule. For the dynamic controller, we set the initial injection probability to $\eta_{\text{init}} = 0.8$, the minimum probability to $\eta_{\text{min}} = 0.0$, and the target validation reward to $R_{\text{target}} = 1.5$. The injection probability $\eta_t$ is then updated according to Eq. 6 as validation performance improves.

The policy is optimized via asymmetric adaptive clipping, using a relaxed upper threshold of $\epsilon_{exp} = 1.0$ for augmented samples to accelerate learning, while enforcing a conservative $\epsilon_{std} = 0.2$ for standard samples and the universal lower bound. For $\text{sim}(a_t, a_t^*)$ in reward, we employ Rouge-L F1 scores normalized to the range $[0, 1]$. The most critical algorithmic hyperparameters for the **Evolution** phase are summarized in Table 5. We employ the **Qwen3-VL-2B-Instruction** and **Qwen3-VL-4B-Instruction** (Yang et al., 2025a) as backbone models. The training framework is based on the EasyR1 codebase (Zheng et al., 2025). All experiments employ BFloat16 (BF16) mixed precision and FlashAttention 2 (Dao, 2023) on 4 NVIDIA A100 (40GB) GPUs.

## A.3. Navigation

We employ the same environmental settings as the *Genesis* phase. Furthermore, the loop continues until a target object is identified or the maximum episode limit of $T_{max} = 50$ steps is reached. In real-world experiments, the *SAGE* Agent was deployed on an NVIDIA RTX 4090 GPU.

*Table 5.* Hyperparameters.

| Parameter | Value |
| --- | --- |
| Global Batch Size | 32 |
| Learning Rate | $1 \times 10^{-6}$ |
| Weight Decay | $1 \times 10^{-2}$ |
| KL Penalty Coefficient | $1 \times 10^{-2}$ |
| Optimizer | AdamW |
| Training Steps | 150 |
| Number of Rollouts | 5 |
| Rollout Temperature | 1.0 |
| Rollout Top_p | 0.99 |
| Max Frames | 4 |
| $\eta_{\text{init}}$ | 0.8 |
| $\eta_{\text{min}}$ | 0.0 |
| $R_{\text{target}}$ | 1.5 |
| Format weight $w_f$ | 0.1 |
| Accuracy weight $w_{acc}$ | 1.0 |
| $\mathcal{P}_{err}$ | 0.5 |

# B. Additional Evaluation Details

### B.1. Baseline Execution Stack Fairness

Since *SAGE* is a planner-assisted high-level navigation method, we separate semantic decision-making from low-level execution in our comparisons. All VLM-based baselines are evaluated with the same frontier generation, occupancy map update, and low-level shortest-path execution stack as *SAGE*.

### B.2. External Model Usage

`Qwen3-VL-2B-Instruction` and `Qwen3-VL-4B-Instruction` are the policy backbones used in *Evolution* and *Navigation*. `Qwen3-VL-Plus` is used only offline during *Genesis* for task and rule synthesis, where higher semantic quality is preferred and the cost is amortized over the generated dataset. `Qwen3-235B-A22B` is used only as an offline A-EQA judge and is not used for training or deployment. Table 6 summarizes the model average usage across stages.

*Table 6.* **Model usage and cost across stages.** Qwen3-VL-Plus is used only for offline Genesis, Qwen3-VL-2B serves as the local backbone in Evolution and Navigation, and Qwen3-235B-A22B is used only for A-EQA answer evaluation. Under the same navigation setting, our method requires fewer VLM calls and lower wall-clock time than 3D-Mem. "/" denotes not applicable.

| Stage | Usage | | Cost (Avg.) | | | |
| | Model | Hardware | Calls | Time | Peak Mem | Tokens |
|---|---|---|---|---|---|---|
| ***Offline Synthesis*** | | | | | | |
| Genesis-HM3D | Qwen3-VL-Plus | Remote API | 52.1k | 16.2h | / | 17.8M |
| Genesis-InteriorGS | Qwen3-VL-Plus | Remote API | 56.1k | 17.5h | / | 19.7M |
| ***Deployment*** | | | | | | |
| Evolution | Qwen3-VL-2B | Local GPUs | / | 29.3h | 38.2 GB/GPU | / |
| Navigation (Ours) | Qwen3-VL-2B | Local GPUs | 9.9k | 25.0h | 11.0GB | 2.5M |
| Navigation (3D-Mem) | Qwen3-VL-2B | Local GPUs | 11.7k | 34.8h | 11.0GB | 3.0M |
| ***Evaluation Only*** | | | | | | |
| A-EQA Eval | Qwen3-235B-A22B | Remote API | 138 | 1.7 min | / | 137.6K |

### B.3. Reliability of Experience Rules

To assess the reliability of VLM-generated IF-THEN experience rules, we manually audited 200 randomly sampled rules across HM3D and InteriorGS on both A-EQA and GOAT-Bench splits. As shown in Table 7, only 10-18% of sampled rules are incorrect or misleading, while most rules are correct or partially correct. This indicates that the retrieved experience bank remains useful because most rules provide usable or benign guidance in aggregate.

*Table 7.* Manual audit of IF-THEN experience rules. Values are percentages.

| Source / Split | Sampled Rules | Correct | Partially Correct | Incorrect |
|---|---|---|---|---|
| HM3D A-EQA | 50 | 42 | 40 | 18 |
| HM3D GOAT-Bench | 50 | 38 | 48 | 14 |
| InteriorGS A-EQA | 50 | 46 | 44 | 10 |
| InteriorGS GOAT-Bench | 50 | 62 | 22 | 16 |

### B.4. Retrieval Relevance and Annealing Sensitivity

We further evaluate whether the benefit of retrieval comes from semantically relevant experience rather than from appending arbitrary text. The mismatched-experience condition retrieves surface-matching candidates from the experience bank and then selects a semantically divergent one. As shown in Table 8, mismatched experience is substantially worse than semantically matched retrieval, confirming the importance of retrieval relevance.

We also evaluate a late-stage annealing schedule for $\epsilon_{exp}$, where $\epsilon_{exp} = 1.0$ is used for the first 50% of training and then

*Table 8.* Effect of mismatched experience injection. All values are in percent (%).

| Method | A-EQA | | GOAT-Bench | |
|---|---|---|---|---|
| | SR$^\dagger$ | SPL$^\dagger$ | SR | SPL |
| Mismatched Exp. | 44.88 | 27.36 | 49.47 | 29.25 |
| *SAGE* | **53.21** | **37.07** | **56.69** | **38.90** |

linearly annealed to 0.3. Table 9 shows that annealing slightly improves A-EQA SPL but reduces SR and GOAT-Bench performance. Therefore, we use the fixed $\epsilon_{exp} = 1.0$ setting as the default.

*Table 9.* Sensitivity to $\epsilon_{exp}$ schedules. All values are in percent (%).

| Schedule | A-EQA | | GOAT-Bench | |
|---|---|---|---|---|
| | SR$^\dagger$ | SPL$^\dagger$ | SR | SPL |
| Fixed $\epsilon_{exp} = 1.0$ | **53.21** | 37.07 | **56.69** | **38.90** |
| Fixed $\epsilon_{exp} = 0.3$ | 46.23 | 29.47 | 49.60 | 28.52 |
| Annealed $1.0 \rightarrow 0.3$ | 51.62 | **38.21** | 55.24 | 37.82 |

## C. Randomization and Reproducibility Control

During the *Evolution* phase, the sampling of the Bernoulli mask $m$ for hybrid training is controlled by a seeded random number generator to ensure training run comparability. To ensure the reliability of our findings, we conducted all training and evaluation runs across 3 independent random seeds (Seeds: 1, 42, 77). As shown in Table 10, the standard deviation (STD) for SR$^\dagger$ across seeds is consistently low ($< 0.5\%$).

In real-world experiments, we standardized external variables to minimize variance. Specifically, all experiments were conducted under constant artificial illumination to negate daylight variations. For object-finding tasks, target objects were reset to valid semantic positions with slight pose variations ($\pm 10°$ rotation) between trials to test robustness without altering task difficulty. We provide a detailed analysis in Appendix J.

*Table 10.* Main Results with Standard Deviation on A-EQA (Supplement to Table 3).

| Method | SR$^\dagger$ (Mean $\pm$ Std) | SPL$^\dagger$ (Mean $\pm$ Std) |
|---|---|---|
| *Qwen3-VL-2B* | $43.51_{\pm 0.26}$ | $27.53_{\pm 0.19}$ |
| $+C_{ret}$ | $46.47_{\pm 0.30}$ | $30.72_{\pm 0.29}$ |
| +Task | $50.71_{\pm 0.34}$ | $33.68_{\pm 0.26}$ |
| +Task+Exp | $51.42_{\pm 0.32}$ | $34.67_{\pm 0.22}$ |
| +Task+Exp+AAC | $51.88_{\pm 0.31}$ | $36.29_{\pm 0.17}$ |
| *SAGE* (Qwen3-2B) | $53.21_{\pm 0.35}$ | $37.07_{\pm 0.28}$ |
| *Qwen3-VL-4B* | $54.13_{\pm 0.27}$ | $36.20_{\pm 0.29}$ |
| $+C_{ret}$ | $55.87_{\pm 0.35}$ | $40.03_{\pm 0.34}$ |
| +Task | $57.14_{\pm 0.31}$ | $45.70_{\pm 0.26}$ |
| +Task+Exp | $58.42_{\pm 0.43}$ | $45.74_{\pm 0.37}$ |
| +Task+Exp+AAC | $59.03_{\pm 0.28}$ | $46.67_{\pm 0.22}$ |
| *SAGE* (Qwen3-4B) | $60.16_{\pm 0.41}$ | $47.21_{\pm 0.33}$ |

## D. Related Work

### D.1. Reasoning-based Embodied Navigation

Embodied navigation requires robots following instructions to execute high-level navigation tasks within visual environments (Khanna et al., 2024; Bar et al., 2025; Han et al., 2025; Xue et al., 2025). Recent research has established two representative task categories: (1) Target-driven navigation (Chaplot et al., 2020; Zhou et al., 2023; Yu et al., 2023; Lei

et al., 2024; Guo et al., 2025), which requires the agent to locate distant objects in unseen environments based on concise instructions or instance images; and (2) Q&A-driven navigation (Cheng et al., 2024a; Saxena et al., 2024; Jiang et al., 2025; Zhai et al., 2025a), which extends the paradigm into interactive scenarios where the agent must explore the scene to comprehend user intent and provide grounded responses. Building on these advancements, recent studies (Zheng et al., 2024; Li et al., 2025a; Qiao et al., 2025) have focused on leveraging VLM to develop general-purpose navigation agents. However, these approaches rely heavily on large, accurately labeled datasets and diverse training data that is scarce and requires dynamic adaptation. This has spurred researchers to explore self-evolving agents, where agents that autonomously acquire data, engage in self-reflective evaluation, and iteratively update themselves.

### D.2. Self-Evolving Agents

Self-evolving agents are capable of achieving autonomous lifelong learning through an iterative loop of experience acquisition, self-assessment, and model updating, all with minimal human intervention (Fang et al., 2025). Substantial strides in agent self-evolution have been witnessed in domains such as logical reasoning (Zhang et al., 2024; Zhao et al., 2025) and tool utilization (Schick et al., 2023; Chen et al., 2023; Hu et al., 2024; Zhai et al., 2025b). Furthermore, recent applications in robotics (Ma et al., 2023; Li et al., 2024; Dong et al., 2025; Li et al., 2025b) validate that this paradigm can autonomously optimize reward functions and control policies via intrinsic feedback. However, within embodied navigation, deploying complex sampling strategies often proves impractical due to the prohibitive trial-and-error costs and exploration inefficiency associated with real-world long-horizon tasks (Wang et al., 2024; Qureshi et al., 2025). A critical obstacle remains the scarcity of structured alignment data required to bridge the gap between high-level intent and low-level robot control. In *SAGE*, the agent mitigates reliance on extensive real-world interaction data and human supervision by leveraging physically grounded abstract experiences, which are autonomously synthesized within a physics-grounded sandbox.

## E. Details for Sandbox Setup

### E.1. InteriorGS Setup

For InteriorGS, we employ a procedural generation approach to facilitate autonomous robot exploration. Specifically, the geometric structure of the sandbox environment is formulated as an occupancy grid map $\mathcal{M}$. To guarantee collision-free trajectories that adhere to rigid-body dynamic constraints, the Euclidean Distance Transform is first applied to compute the distance $d(p)$ from each free-space pixel $p$ to the nearest obstacle. A safe configuration space $\mathcal{C}_{safe} = \{p \in \mathcal{M} \mid d(p) \geq \delta\}$ is then defined based on a minimum safety threshold $\delta = 5$. Within this valid region, start $s_{start}$ and goal $s_{goal}$ points are randomly sampled, followed by an A* search to determine the shortest trajectory $\mathcal{T}$. Data diversity and validity are maintained by filtering trajectories based on length (less than 20 meters) and performing discrete sampling $T$ ($2 \leq T < 10$) on $\mathcal{T}$. Subsequently, heading angles are derived from the tangent direction of adjacent nodes, yielding a smooth trajectory $\tau^* = \{(x_t, y_t, \theta_t)\}_{t=0}^{T}$ that encompasses both position and orientation. High-resolution RGB-D observations are simulated by mapping these trajectories into the 3D world coordinate system. At each keypoint along $\tau^*$, a simulated camera is positioned 1.5m above the ground with a $120°$ HFOV. Finally, synchronized RGB observations are synthesized from three egocentric views (forward, left, and right), denoted as $\Omega_t = \{\mathcal{R}((x_t, y_t, \theta_t), \zeta) \mid \zeta \in \{0, -\frac{2\pi}{3}, \frac{2\pi}{3}\}\}$, using the differentiable Gaussian rasterizer $\mathcal{R}$.

### E.2. HM3D Setup

Considering that the A-EQA and GOAT-Bench tasks utilize the HM3D dataset, we removed duplicate scenes to ensure training and validation trajectories did not overlap. We selected 698 high-quality scenes from the dataset as our data source. For the HM3D dataset, trajectory generation is modeled as a constrained path planning problem over a continuous navigation mesh. To achieve trajectory diversity, we leverage the Habitat simulator's underlying Pathfinder (Szot et al., 2021). Specifically, consistent with the InteriorGS setup, a start point $s_{start}$ and a goal point $s_{goal}$ are uniformly sampled within the traversable space of each scene, followed by the computation of the geodesic shortest trajectory $\mathcal{T} = \{s_1, s_2, ..., s_T\}$. To ensure the validity and non-triviality of the trajectories for task learning, a path length constraint (specifically, less than 20 meters) is imposed; samples violating this threshold or failing the planning process are discarded. For each keypoint $(x_t, y_t)$, the heading angle $\theta_t = \arctan2(\Delta y, \Delta x)$ is derived from the tangent direction toward the subsequent node $(x_{t+1}, y_{t+1})$, with the terminal point $(x_T, y_T)$ inheriting the orientation of its predecessor. Finally, the simulated camera is uniformly positioned 1.5m above the ground with a $120°$ HFOV at each keypoint, synthesizing synchronized RGB observations from three egocentric views (forward, left, and right).

### E.3. Details for Goal-Conditioned Embodied Question Answering Task Types

Our objective is to generate a massive scale of diverse, practical, and complex Goal-Conditioned Embodied Question Answering tasks. We initiate the process by extracting the forward intermediate observation of the trajectory endpoint. The object detection model is then employed to acquire bounding boxes, spatial coordinates, and category labels for each object, from which we sample specific object instances. Subsequently, we feed these object attributes and corresponding visual contexts, alongside few-shot Q&A examples, into `Qwen3-VL-Plus` to simulate natural domestic dialogue. Guided by specific prompts, we instruct `Qwen3-VL-Plus` to randomly synthesize tasks across eight distinct categories designed to encompass comprehensive cognitive capabilities: Object Recognition, Object Localization, Attribute Recognition, Object State Recognition, Counting, World Knowledge, Spatial Understanding, and Functional Reasoning. The resulting answers are formatted as either open-ended or multiple-choice responses, facilitating the assessment of diverse agent capabilities. The following are some examples:

**Attribute Recognition**
*Question:* Is there space on the wooden bench to the left for placing a laptop?
*Answer:* Yes, there is clear space on the wooden bench to the left for placing a laptop.

**Counting**
*Question:* How many of the listed objects — vent, toaster, picture — are positioned to the right of the vent?
*Answer:* Two.

**Functional Reasoning**
*Question:* I want to relax with a book on the couch, where should I place the book so it's easily reachable while sitting?
*Answer:* Place the book on the coffee table, which is directly in front of the couch.

**Object Localization**
*Question:* Where is the potted plant located?
*Answer:* At the far end of the hallway, visible through the open doorway.

**Object Recognition**
*Question:* What is the green plant in the pot near the bench?
*Answer:* A potted plant.

**Object State Recognition**
*Question:* Is the sofa chair positioned above the cabinet in the room?
*Answer:* Yes, the sofa chair is positioned above the cabinet.

**Spatial Understanding**
*Question:* What is the relative position of the couch and the sofa chair?
*Answer:* The couch is to the left of the sofa chair.

**World Knowledge**
*Question:* What type of furniture is visible in the hallway leading to the bedroom?
*Answer:* A couch is visible in the hallway.

### E.4. Trajectory Verification

To maintain generated data quality, we designed the trajectory validator to filter out low-quality data through a multi-stage rejection sampling strategy. For each scene, the system first ensures visual validity by analyzing semantic scene graph labels from the forward view of the trajectory endpoint. Specifically, trajectories are rejected if they fail to capture any identifiable 2D targets or terminate at meaningless objects like walls, ensuring the agent focuses on unique and interactive goals.

Once a trajectory satisfies these physical and visual constraints, we validate the consistency of output templates to ensure syntactic integrity of associated output data. To prevent hallucinations or structural inconsistencies, we strictly constrain

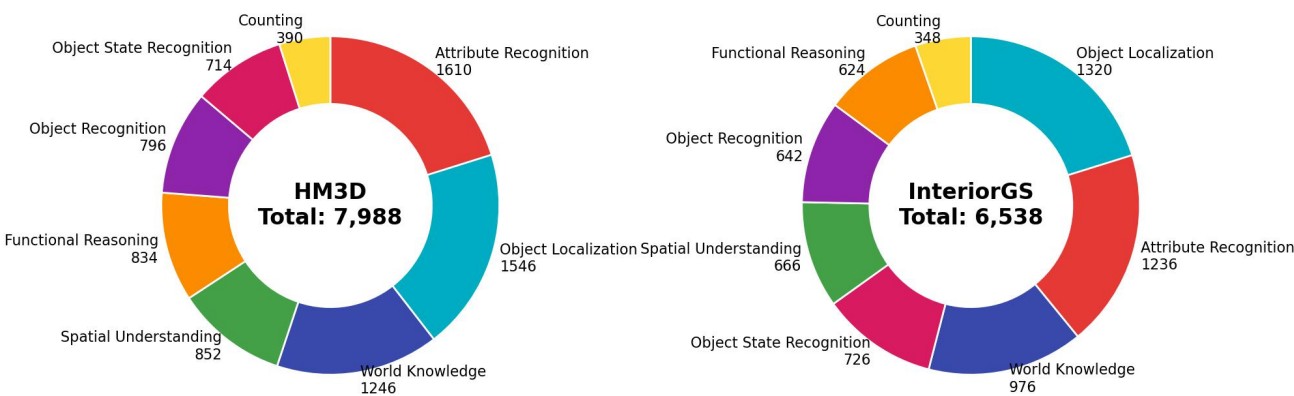

*Figure 6.* Distribution of sandbox task categories.

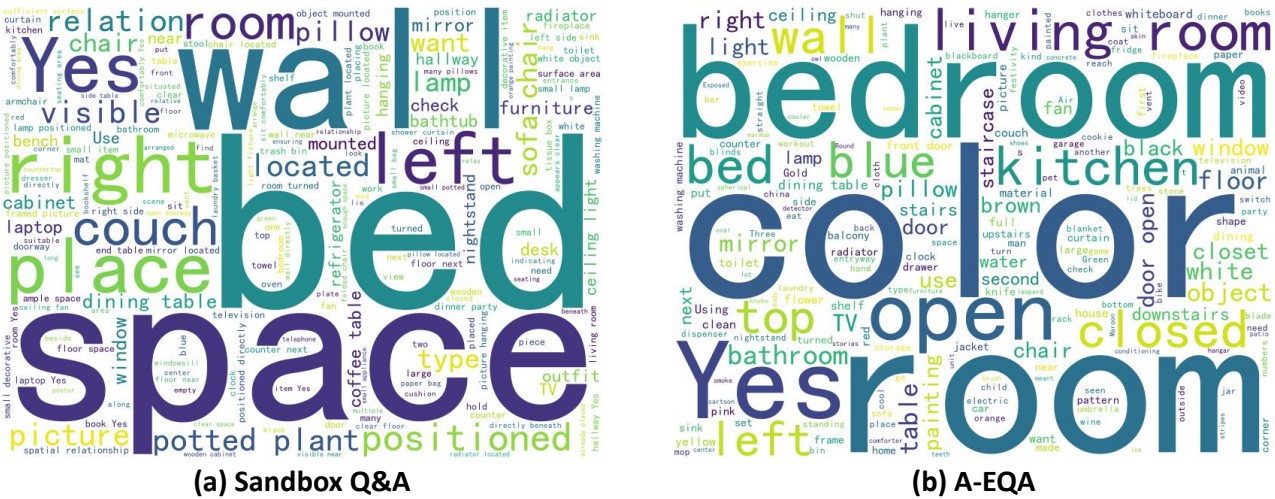

**(a) Sandbox Q&A**             **(b) A-EQA**

*Figure 7.* Visualization of the word cloud.

rules using regular expressions. The entire trajectory is discarded if the generated output fails to match the required templates, specifically the defined Task/Question/Answer format for EQA or the logical IF-AND-THEN structure for experience descriptions. Only candidates that successfully pass these rigorous formatting checks without raising parsing exceptions are serialized and counted towards the final sandbox trajectory.

For verification purposes, our experiments collected 14,526 valid trajectories through the aforementioned procedural rules: 7,988 from HM3D and 6,538 from InteriorGS, with category distributions shown in Figure 6.

Additionally, we analyzed word clouds for the trajectory QA dataset and A-EQA tasks. As shown in Figure 7, we comparatively analyze the word clouds for the A-EQA baseline and our self-evolved sandbox trajectory QA pairs. For A-EQA, the vocabulary is dominated by low-level visual attributes and coarse scene labels, with "color", "bedroom", and "living room" appearing most frequently, indicating a focus on static recognition tasks. The sandbox dataset reveals a shift towards complex spatial reasoning and affordance understanding. Keywords like "space", "place", and "positioned" take prominence, implying that the generated queries require agents to analyze geometric availability and object relationships. Moreover, directional terms such as "left", "right", and "near" recur frequently alongside concrete obstacles (e.g., "bed", "couch"), demonstrating that our self-evolving pipeline effectively synthesizes high-quality tasks that challenge an agent's ability to comprehend fine-grained spatial context rather than simple visual attributes.

Since the GOAT-Bench task itself lacks Q&A pairs, we manually constructed questions for different goal-oriented navigation scenarios, as shown in the Table 11.

---

**Algorithm 1** Workflow of *Navigation*

---

**Require:** Evaluation Dataset $\mathcal{D}_{eval}$, Experience Database $\mathcal{D}_{exp}$ (from Genesis), Policy $\pi_\theta$ (from Evolution, Eq. 11)
**Require:** Max steps $T_{max}$, Step limit $\delta_{max}$
1: **for** each task instance $n \in \mathcal{D}_{eval}$ **do**
2:     **Initialize:** Robot pose $\xi_0$, Query $I_{query}$
3:     **Initialize Buffers:** Memory $\mathcal{M}_0 \leftarrow \emptyset$, Frontier $\mathcal{F}_0 \leftarrow \emptyset$
4:     done $\leftarrow$ false
5:     **for** $t = 0$ to $T_{max}$ **do**
6:         **Perception:** Obtain RGB-D observation $O_t$ at pose $\xi_t$
7:         *// 1. Structured Perception Update (Section 3.3)*
8:         Update 3D Scene Graph and Occupancy Map using $O_t$
9:         Update Memory Buffer: $\mathcal{M}_t \leftarrow \text{UpdateMemory}(\mathcal{M}_{t-1}, O_t)$
10:        Update Frontier Buffer: $\mathcal{F}_t \leftarrow \text{ComputeFrontiers}(\text{OccupancyMap})$
11:        *// 2. State Representation Construction (Eq. 12)*
12:        Construct state $\mathcal{R}_t = \{I_{query}, \mathcal{M}_t, \mathcal{F}_t\}$
13:        *// 3. Experience Augmentation via Retrieval*
14:        Query $\mathcal{D}_{exp}$ with $O_t$ to retrieve relevant priors:
15:        $C_{ret} \leftarrow \text{Retrieve}(\mathcal{D}_{exp}, O_t, I_{query})$
16:        *// 4. Policy Decision via Evolved VLM (Eq. 13)*
17:        Select optimal sub-goal $a_t^*$ from action space $\mathcal{A} = \mathcal{F}_t \cup \mathcal{M}_t$:
18:        $a_t^* = \text{argmax}_{a \in \mathcal{A}} \pi_\theta(a \mid \mathcal{R}_t, C_{ret})$
19:        **if** $a_t^* \in \mathcal{M}_t$ **then**
20:           *// Target Identification / QA*
21:           Generate answer or navigate to target object
22:           done $\leftarrow$ true
23:           **break**
24:        **else if** $a_t^* \in \mathcal{F}_t$ **then**
25:           *// Exploration via Geometric Planner*
26:           Plan shortest path $\tau_{min}$ to frontier node $a_t^*$
27:           $\xi_{t+1} \leftarrow \text{ExecutePath}(\tau_{min}, \text{limit} = \delta_{max})$
28:        **else**
29:           **break** *// Invalid decision*
30:        **end if**
31:     **end for**
32: **end for**

---

## F. Theoretical Analysis for Asymmetric Adaptive Clipping

In this section, we provide a theoretical justification for the *Asymmetric Adaptive Clipping* (AAC) mechanism introduced in the *Evolution* phase. We demonstrate that AAC preserves the fundamental trust-region guarantees of PPO/GRPO for stability while asymptotically approaching a Weighted Behavior Cloning objective.

### F.1. Preliminaries

The standard GRPO (Shao et al., 2024) objective approximates the PPO surrogate by estimating the baseline from group scores rather than a value function. For a group of outputs $\{o_i\}_{i=1}^G$ sampled from $\pi_{\theta_{old}}$, the objective is:

$$J_{GRPO}(\theta) = \mathbb{E}\left[\frac{1}{G}\sum_{i=1}^{G}\frac{1}{|o_i|}\sum_{t=1}^{|o_i|}\min\left(\rho_{i,t}(\theta)\hat{A}_{i,t}, \text{clip}(\rho_{i,t}(\theta), 1-\epsilon, 1+\epsilon)\hat{A}_{i,t}\right) - \beta\mathbb{D}_{KL}\right], \quad (16)$$

where $\rho_{i,t} = \frac{\pi_\theta(o_{i,t}|q, o_{i,<t})}{\pi_{\theta_{old}}(o_{i,t}|q, o_{i,<t})}$ is the probability ratio, and $\hat{A}_{i,t}$ is the group-relative advantage. The clipping mechanism $\text{clip}(\rho_{i,t}(\theta), 1-\epsilon, 1+\epsilon)$ ensures the new policy $\pi_\theta$ does not deviate excessively from $\pi_{\theta_{old}}$, preventing performance collapse.

*Table 11.* Goal Categories and Examples

| Goal Category | Question | Example |
|---|---|---|
| Object | Where can I find {goal_category}? | Where can I find piano? |
| Description | Where can I find the object exactly described as the {goal_description}? | Where can I find the object exactly described as the 'the grand piano located to the right and below the cabinet in a log cabin.'? |
| Image | Where can I find the exact object captured at the center of the following image? |  |

In *SAGE*, we modify the clipping function to be asymmetric based on the sample source mask $m_i \in \{0, 1\}$, where $m_i = 1$ denotes an augmented experience sample. The asymmetric boundaries are defined as:

$$\epsilon_{down} = \epsilon_{std}, \tag{17}$$

$$\epsilon_{up}(m_i) = \begin{cases} \epsilon_{exp} & \text{if } m_i = 1 \\ \epsilon_{std} & \text{if } m_i = 0 \end{cases}, \tag{18}$$

with $\epsilon_{exp} \gg \epsilon_{std}$.

## F.2. Preservation of Policy Safety Lower Bound

**Theorem 1.** *For any sample trajectory with negative advantage ($\hat{A}_{i,t} < 0$), the AAC objective restricts the policy update, regardless of the upper clipping threshold $\epsilon_{exp}$.*

**Proof.** Consider the GRPO surrogate objective $L(\theta)$ for a single token $t$ and sample $i$:

$$L_{i,t}(\theta) = \min(\rho_{i,t}(\theta)\hat{A}_{i,t}, \text{clip}(\rho_{i,t}(\theta), 1 - \epsilon_{down}, 1 + \epsilon_{up})\hat{A}_{i,t}). \tag{19}$$

When $\hat{A}_{i,t}$ is negative, the $\min$ operator selects the greater of the two terms. The constraint becomes the lower bound of the clip function:

$$L_{i,t}(\theta) = \max(\rho_{i,t}(\theta)\hat{A}_{i,t}, (1 - \epsilon_{std})\hat{A}_{i,t}). \tag{20}$$

Since $\hat{A}_{i,t} < 0$, the maximum operator selects the term that is less negative. The penalty for "bad" actions is bounded by $1 - \epsilon_{std}$.

**Conclusion.** AAC retains the exact same lower bound as standard GRPO. The policy is strictly protected against catastrophic probability collapse for sub-optimal actions, regardless of whether they come from retrieved experience or exploration.

## F.3. Preservation of Policy Safety Upper Bound.

**Theorem 2.** *For augmented samples with positive advantage ($\hat{A}_{i,t} > 0$), If $\epsilon_{exp} \to \infty$ is permitted, monotonic policy improvement cannot be guaranteed.*

**Proof.** When the advantage is positive, the objective seeks to increase $\rho_{i,t}(\theta) > 1$. For Standard Samples ($m_i = 0$), the upper bound is $1 + \epsilon_{std}$. For Augmented Samples ($m_i = 1$), the upper bound is relaxed to $1 + \epsilon_{exp}$ (where $\epsilon_{exp} > \epsilon_{std}$):

$$L_{aug}^{CLIP}(\theta) = \min(\rho_{i,t}(\theta)\hat{A}_{i,t}, (1 + \epsilon_{exp})\hat{A}_{i,t}). \tag{21}$$

By relaxing the upper bound, we allow the policy to take larger gradient steps toward the retrieved trajectory. Consider the SAGE objective (Eq. 11) for an augmented sample with $\hat{A}_{i,t} > 0$. If we set $\epsilon_{exp} \to \infty$, the objective simplifies to:

$$J(\theta) = \rho_{i,t}(\theta)\hat{A}_{i,t} - \beta D_{KL}, \tag{22}$$

*Table 12.* **Comparison with baselines on A-EQA by Question Categories.** "CG" denotes ConceptGraphs, "SVM" denotes Sparse Voxel Map. Methods with * are reported from OpenEQA (Majumdar et al., 2024) and 3D-Mem (Yang et al., 2025b). *SAGE*$^{\dagger}$ represents full-set evaluation. All values are in percent (%). **Bolded numbers** highlight the best results.

| Method | VLM/LLM | Object Recognition | | Object Localization | | Attribute Recognition | | Spatial Understanding | | Object State Recognition | | Functional Reasoning | | World Knowledge | | Overall | |
|---|---|---|---|---|---|---|---|---|---|---|---|---|---|---|---|---|---|
| | | SR$^{\dagger}$ | SPL$^{\dagger}$ | SR$^{\dagger}$ | SPL$^{\dagger}$ | SR$^{\dagger}$ | SPL$^{\dagger}$ | SR$^{\dagger}$ | SPL$^{\dagger}$ | SR$^{\dagger}$ | SPL$^{\dagger}$ | SR$^{\dagger}$ | SPL$^{\dagger}$ | SR$^{\dagger}$ | SPL$^{\dagger}$ | SR$^{\dagger}$ | SPL$^{\dagger}$ |
| *Human Agent** | | 89.7 | - | 72.8 | - | 85.4 | - | 84.8 | - | 97.8 | - | 78.9 | - | 88.5 | - | 85.1 | - |
| ***Blind LLMs*** | | | | | | | | | | | | | | | | | |
| GPT-4* | GPT-4 | 25.3 | - | 28.4 | - | 27.3 | - | 37.7 | - | 47.2 | - | 54.2 | - | 29.5 | - | 35.5 | - |
| GPT-4o* | GPT-4o | 22.0 | - | 25.0 | - | 27.3 | - | 40.8 | - | 50.9 | - | 61.8 | - | 38.4 | - | 35.9 | - |
| Qwen3-Max | Qwen3-Max | 10.0 | - | 28.6 | - | 26.5 | - | 42.1 | - | 40.7 | - | 54.4 | - | 26.8 | - | 31.0 | - |
| Qwen3-235b-a22b | Qwen3-235b-a22b | 22.0 | - | 35.0 | - | 25.8 | - | 39.5 | - | 65.7 | - | 57.4 | - | 26.8 | - | 37.4 | - |
| ***RL Paradigm*** | | | | | | | | | | | | | | | | | |
| SenseAct-NN Skill Chain | GPT-4o | 15.7 | 6.4 | 24.6 | 12.6 | 23.7 | 9.9 | 36.1 | 15.4 | 39.2 | 20.2 | 45.7 | 21.5 | 22.4 | 14.7 | 24.7 | 13.3 |
| SenseAct-NN Monolithic | GPT-4o | 12.3 | 4.5 | 23.7 | 8.8 | 20.3 | 9.7 | 25.7 | 11.6 | 37.3 | 18.3 | 32.8 | 18.4 | 19.6 | 8.7 | 20.6 | 10.1 |
| ***LLM w/ Captions*** | | | | | | | | | | | | | | | | | |
| CG Scene-Graph Captions* | GPT-4 | 25.3 | - | 16.5 | - | 29.2 | - | 37.0 | - | 52.2 | - | 46.8 | - | 37.8 | - | 34.4 | 6.5 |
| SVM Scene-Graph Captions* | GPT-4 | 29.0 | - | 17.2 | - | 31.5 | - | 31.5 | - | 54.2 | - | 39.8 | - | 38.9 | - | 34.2 | 6.4 |
| LLaVA-1.5 Frame Captions* | GPT-4 | 25.0 | - | 24.0 | - | 34.1 | - | 34.4 | - | 56.9 | - | 53.5 | - | 40.6 | - | 38.1 | 7.0 |
| Multi-Frame* | GPT-4V | 34.0 | - | 34.3 | - | 51.5 | - | 39.5 | - | 51.9 | - | 45.6 | - | 36.6 | - | 41.8 | 7.5 |
| ***VLM Paradigm*** | | | | | | | | | | | | | | | | | |
| Explore-EQA* | GPT-4o | 44.0 | 19.6 | 37.1 | 29.6 | 55.3 | 36.0 | 42.1 | 6.6 | 46.3 | 9.2 | 63.2 | 35.7 | 45.5 | 22.0 | 46.9 | 23.4 |
| 3D-Mem* | GPT-4o | 49.0 | 45.2 | 48.6 | 41.3 | 47.7 | 38.6 | 43.4 | 33.3 | 69.4 | 50.3 | 64.7 | 47.2 | 49.1 | 38.9 | 52.6 | 42.0 |
| 3D-Mem | Qwen3-VL-2B | 34.8 | 16.0 | 44.9 | 22.2 | 47.1 | 25.2 | 52.6 | 22.1 | 48.0 | 16.9 | 34.7 | 6.1 | 45.5 | 20.5 | 44.3 | 19.4 |
| *SAGE* (Ours) | Qwen3-VL-2B | 51.2 | 41.7 | 49.4 | 31.2 | **60.5** | **47.5** | 52.1 | 28.7 | 64.3 | 40.7 | 43.2 | 30.6 | 47.3 | 34.1 | 53.2 | 37.1 |
| *SAGE*$^{\dagger}$ | Qwen3-VL-2B | 47.4 | 34.2 | 49.0 | 29.4 | 54.7 | 35.4 | 55.5 | 38.5 | 67.8 | 44.9 | 56.8 | 42.7 | 50.2 | 33.4 | 54.1 | 36.2 |
| *SAGE* | Qwen3-VL-4B | **61.8** | **51.5** | **59.4** | **47.4** | 54.1 | 40.8 | **53.2** | **40.0** | 69.6 | 54.4 | 65.3 | 49.6 | 59.3 | 47.1 | **60.2** | **47.2** |

where $\rho_t = \pi_\theta / \pi_{\theta_{old}}$. Although the KL penalty term $-\beta D_{KL}$ theoretically discourages infinite updates, the linear gain term $\rho_{i,t} \hat{A}_{i,t}$ can dominate locally if $\hat{A}_{i,t}$ is large.

**Conclusion.** Therefore, $\epsilon_{exp}$ must be finite. It acts as a relaxed but necessary trust region.

### F.4. Variance Control via Homogeneous Grouping

A potential risk of relaxed clipping is gradient variance. However, *SAGE* leverages the Group Relative nature of GRPO to mitigate this. Since normalization is conducted within homogeneous groups, a reasonably relaxed $\epsilon_{exp}$ ensures that the magnitude of $\hat{A}_{i,t}$ in augmented samples does not increase disproportionately relative to that of standard samples. As demonstrated in Section 4.4, we empirically verify that even with $\epsilon_{exp} = 1.0$, the gradients derived from aggressive AAC updates do not numerically overwhelm the conservative standard update gradients, thereby preserving the training stability of the base model.

## G. Implementation Details for *Navigation* Workflow

During navigation, the robot performs panoramic sampling using a camera with a $120°$ HFOV and a resolution of $1280 \times 1280$. In the initial step, a total of 7 views are acquired by capturing 6 additional perspectives at $40°$ intervals; for subsequent steps, 3 views are obtained via 2 additional perspectives at $60°$ intervals. Observations within a depth threshold of 1.7m are fused into a voxel-based occupancy map with a 0.1m resolution, while the scene graph and memory are simultaneously updated. Considering that the Memory Buffer $\mathcal{M}_t$ contains numerous images unrelated to the task, we employ the Pre-filtering mechanism (Yang et al., 2025b) to filter $\mathcal{M}_t$. Subsequently, the VLM selects a target from either the Frontier Buffer $\mathcal{F}_t$ or the Memory Buffer $\mathcal{M}_t$ (with prompt images resized to $256 \times 256$). The planner converts this target into a reachable navigation waypoint and advances the robot with a maximum step size of 1.0m. Specifically, for Memory-type targets, the robot halts at an observation point approximately 0.75m from the target for verification. A radius of 0.7m is continuously marked as explored until the snapshot target is reached or the maximum step count is exhausted. The overall workflow is illustrated in Algorithm 1.

## H. Detailed Results

**A-EQA Baselines.** We benchmark *SAGE* against a diverse of RL and VLM baselines. For the RL paradigm, we adopt the SenseAct-NN baselines from GOAT-Bench, wherein GPT-4o is invoked to answer questions based on the current viewpoint

*Table 13.* **Comparison with baselines on GOAT-Bench by Task Categories.** Methods with * are reported from 3D-Mem. *SAGE*[†] represents full-set evaluation. All values are in percent (%). **Bolded numbers** highlight the best open-source results.

| Method | VLM/LLM | Object Category | | Language | | Image | | Overall | |
|---|---|---|---|---|---|---|---|---|---|
| | | SR | SPL | SR | SPL | SR | SPL | SR | SPL |
| *RL Paradigm* | | | | | | | | | |
| SenseAct-NN Skill Chain | - | 25.8 | 17.0 | 16.3 | 7.4 | 42.2 | 18.0 | 29.5 | 11.3 |
| SenseAct-NN Monolithic | - | 25.7 | 13.7 | 12.6 | 6.5 | 11.7 | 7.3 | 12.3 | 6.8 |
| Modular GOAT | - | 29.4 | 17.0 | 21.5 | 16.2 | 27.9 | 19.5 | 24.9 | 17.2 |
| *Closed-Source VLMs* | | | | | | | | | |
| Explore-EQA* | GPT-4o | 64.7 | 48.4 | 42.9 | 22.7 | 56.8 | 41.8 | 55.0 | 37.9 |
| 3D-Mem* | GPT-4o | 79.2 | 55.8 | 61.9 | 46.0 | 65.2 | 44.2 | 69.1 | 48.9 |
| *Open-Source VLMs* | | | | | | | | | |
| 3D-Mem | Qwen3-VL-2B | 53.5 | 21.7 | 45.1 | 14.9 | 39.8 | 24.4 | 46.4 | 20.3 |
| 3D-Mem* | LLaVA-7B | 62.6 | 33.3 | 49.5 | 31.7 | 35.2 | 22.7 | 49.6 | 29.4 |
| *SAGE* (Ours) | Qwen3-VL-2B | 70.8 | 43.1 | 47.7 | **33.3** | 48.6 | 40.1 | 56.7 | 38.9 |
| *SAGE*[†] | Qwen3-VL-2B | 68.0 | 41.4 | 45.7 | 31.9 | 43.6 | 36.0 | 53.3 | 36.6 |
| *SAGE* | Qwen3-VL-4B | **73.7** | **50.9** | **54.6** | 31.9 | **69.2** | **58.1** | **64.8** | **44.9** |

immediately upon the policy's issuance of a "STOP" action. The LLM w/ Captions paradigm employs frontier-based exploration to accumulate scene memory from image frames. Specifically, this includes CG Scene-Graph Captions, which utilizes ConceptGraphs (CG) to construct a textual scene-graph representation for prompting the LLM; SVM Scene-Graph Captions, which leverage Sparse Voxel Map (SVM) to build a corresponding textual representation; LLaVA-1.5 Frame Captions, which uses LLaVA-1.5 to generate frame-level captions; and Multi-Frame, which directly prompts the VLM with multiple frames. Regarding the VLM paradigm, exploration is terminated the moment the VLM deems the current viewpoint sufficient for resolving the query. Additionally, we incorporate a Blind LLMs baseline to evaluate performance in the complete absence of visual information. Finally, as is customary, due to resource constraints, our main text evaluation is conducted on a subset of A-EQA. For reference, we provide the full-set evaluation *SAGE*[†]. All baselines follow their official implementations.

The quantitative results presented in Table 12 that *SAGE* establishes a new state-of-the-art across all metrics. By effectively bridging high-level reasoning with low-level control, *SAGE* (Qwen3-VL-4B) achieves an overall SR[†] of 60.2% and SR[†] of 47.2%, significantly outperforming the RL paradigm baselines, which struggle to align semantic goals with sparse rewards. Furthermore, *SAGE* surpasses the LLM w/ Captions methods by a wide margin, validating that our direct processing of visual inputs preserves critical spatial details often lost in textual scene-graph abstractions. Most notably, *SAGE* exhibits exceptional data efficiency and reasoning capability compared to the strong VLM paradigm baseline, 3D-Mem. Even our lightweight *SAGE* (Qwen3-VL-2B) model achieves an overall score of 53.2%, surpassing the 3D-Mem agent driven by the significantly larger GPT-4o (52.6%). This result serves as a powerful testament to our Genesis and Evolution, as internalizing sandbox-derived experience priors, a smaller parameter model can develop superior embodied intuition than a closed-source giant relying on zero-shot prompting. Additionally, *SAGE* (4B) dominates in complex reasoning categories such as Spatial Understanding (53.2% vs. 43.4%), confirming that our retrieved experience rules effectively guide the agent through multi-step logical deductions. Finally, the superior SPL indicate that *SAGE* does not merely stumble upon answers via exhaustive search but navigates with purpose, mirroring the optimal behaviors distilled from the sandbox.

**GOAT-Bench Baselines.** Corroborating our findings on A-EQA, *SAGE* demonstrates robust generalization on the GOAT-Bench (Table 13), which places a heavier emphasis on navigation fidelity. Consistent with the trends observed in the A-EQA, *SAGE* establishes a significant lead over the RL paradigm, with our 4B model achieving an overall SR of 64.8% and SPL of 44.9%, effectively tripling the efficiency of Modular GOAT.

In the realm of local models, *SAGE* establishes a dominant lead. Our SAGE (Qwen3-VL-2B) achieves an overall SR of 56.7%, surpassing the equivalent baseline (46.4%) by over 10 percentage points and even outperforming the larger LLaVA-7B. Crucially, the improvement in path efficiency is even more pronounced. *SAGE* (2B) achieves an SPL of 38.9% compared to 3D-Mem's 20.3%. This near-doubling of efficiency validates that our Evolution phase does not merely teach

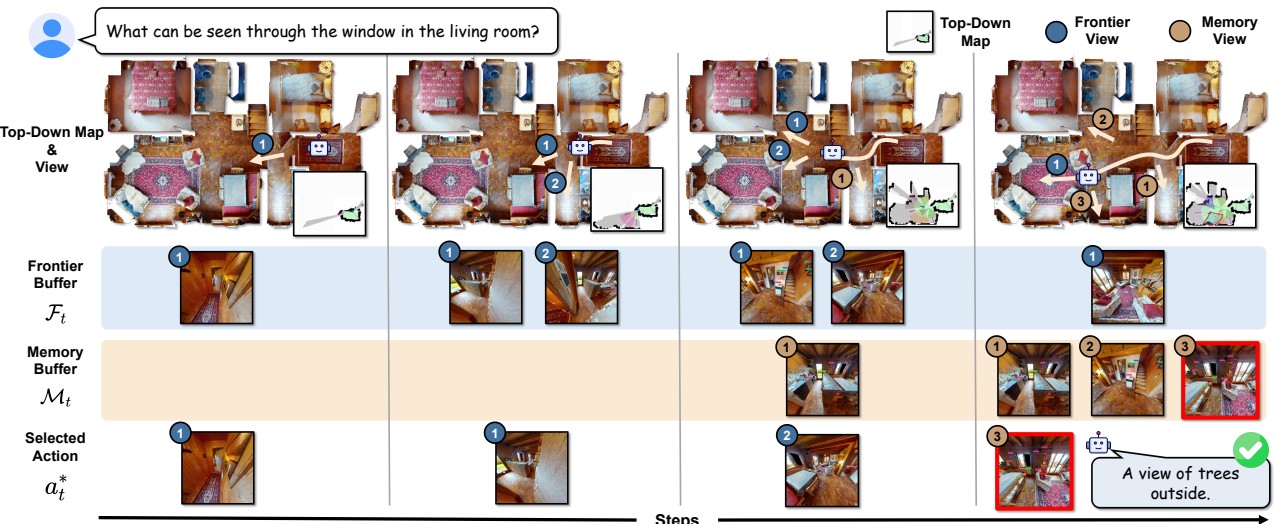

Figure 8. Qualitative example of the *SAGE* agent on A-EQA task. Given a natural language query, the *SAGE* agent maintains a Frontier Buffer $\mathcal{F}_t$ for exploration candidates and a Memory Buffer $\mathcal{M}_t$ for semantic history. (Top) The evolving top-down occupancy map and the robot's trajectory (blue line), showing the progressive exploration of the environment.(Middle) The Frontier Buffer (blue block) stores candidate nodes for exploration, while the Memory Buffer (orange block) aggregates observed semantic landmarks. (Bottom) The Selected Action $a_t^*$ chosen by the VLM policy. In the initial steps, the agent prioritizes exploration by selecting frontier nodes. In the final step, the agent successfully identifies the target view from the Memory Buffer to derive the answer "A view of trees outside.

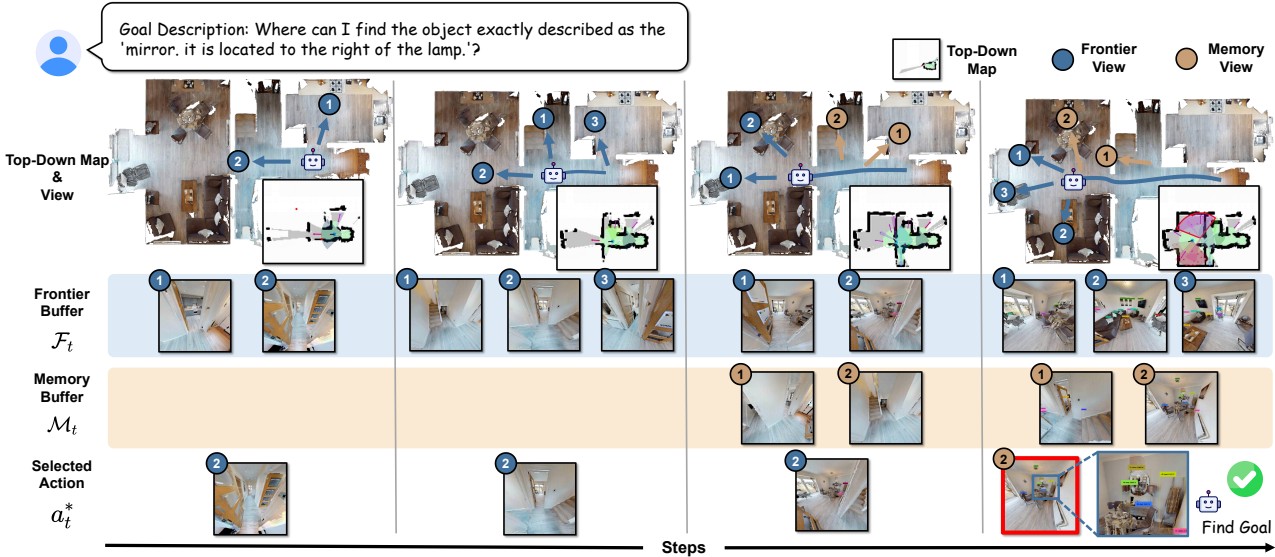

Figure 9. Qualitative example of the *SAGE* agent on GOAT-Bench task. The agent is instructed to find a specific object based on a spatial description. (Top) The robot's trajectory (blue line) navigates through multiple rooms to locate the target. (Middle) The distinct Frontier Buffer $\mathcal{F}_t$ and Memory Buffer $\mathcal{M}_t$. Throughout the steps, the agent actively selects frontier nodes to traverse the hallway and enter the bedroom. (Bottom) The Selected Action $a_t^*$. In the final step, the agent successfully grounds the complex natural language description into visual reality. Upon recognizing the specific mirror positioned to the right of a lamp, it switches strategy to select the corresponding node from the Memory Buffer as the final goal.

the agent what to find, but how to find it optimally, mitigating the erratic exploration often seen in standard VLM agents. Additionally, while GPT-4o performs slightly better, it is closed-source and cloud-dependent. In contrast, as a local model, *SAGE* significantly outperforms comparable models, demonstrating exceptional edge-side usability. This result further validates that self-evolving on high-quality procedural data is a viable path to achieving SOTA performance without relying on massive scale or proprietary APIs.

## I. Qualitative Analysis

### I.1. A-EQA

We provide a detailed qualitative visualization of the A-EQA task decision-making inference process in Figure 8, demonstrating how *SAGE* addresses the query, "What can be seen through the window in the living room?". Initially, as the target object is not within the current field of view, the agent prioritizes active exploration by selecting unvisited candidate nodes from the Frontier Buffer $\mathcal{F}_t$, effectively guiding the robot to navigate through the hallway and into the open living area. As the trajectory extends, the Memory Buffer $\mathcal{M}_t$ incrementally aggregates semantic observations of the environment. In the final step, upon recognizing the target visual cues, the policy shifts its strategy from exploration to exploitation; it correctly identifies and selects the specific frame depicting the window from the Memory Buffer as the optimal action $a_t^*$. This explicit selection allows the agent to ground the high-level textual query into precise visual evidence, ultimately deriving the correct answer, "A view of trees outside," thereby validating *SAGE*'s capability to bridge long-horizon semantic planning with precise low-level control through structured visual abstractions.

### I.2. GOAT-Bench

We provide a detailed qualitative visualization of the GOAT-Bench task decision-making inference process in Figure 9, specifically illustrating the "Goal Description" category. As depicted, the *SAGE* agent initially prioritizes the Frontier Buffer $\mathcal{F}_t$ during steps to explore the unknown environment, efficiently traversing from the corridor into the target room. Upon observing the scene at the final step, the VLM policy transitions to reasoning over the Memory Buffer $\mathcal{M}_t$, where it successfully grounds the complex natural language description into visual reality. By verifying the geometric relationship, the agent distinguishes the correct target from distractors and selects the corresponding memory node as the optimal action $a_t^*$, demonstrating robust capabilities in bridging high-level semantic intent with precise instance-level grounding.

*Table 14.* Computational Cost per Navigation Step (Real-World)

| Component | Hardware / Spec | Metric | Value |
|---|---|---|---|
| VLM Inference | NVIDIA RTX 4090 (24GB) | Latency | $2242_{\pm 1419}$ ms |
| | | VRAM Usage | 18.7 GB (Peak) |
| Top-Down Map Construction | NVIDIA RTX 4090 (24GB) | Latency | $76_{\pm 51}$ ms |
| Object Detection | NVIDIA RTX 4090 (24GB) | Latency | $360_{\pm 229}$ ms |
| Visual Processing | Onboard CPU (Intel(R) Core(TM) Ultra 5 125H) | Latency | $32_{\pm 16}$ ms |
| Network Transmission | WiFi 6 (5GHz) | Latency | $25_{\pm 5}$ ms |
| Path Planning | ROS Navigation Stack | Latency | $< 10$ ms |
| Total Step Latency | End-to-End | Time | $\sim 2.7$ s |

## J. Real-World Deployment

### J.1. Robot Setup

*SAGE* is evaluated on a Qizhi ROS robot equipped with a Kinect v2 RGB-D camera (see schematic in Figure 10). The camera is mounted at a height of 1.5m with a pitch angle of 0 radians. Sensor specifications include a depth resolution of $512 \times 424$ (valid range: 0.5m-6.0m) and an RGB resolution of $1920 \times 1080$. An onboard computer manages data interfacing and velocity control, communicating via WiFi with a remote server to transmit captured images and receive navigation commands. Our model is deployed on a remote server equipped with an NVIDIA RTX 4090 GPU. During navigation, the server communicates via WiFi to receive navigation queries and images captured by the robot. To ensure transmission efficiency, images are compressed to a resolution of $256 \times 256$ prior to transfer. Upon processing the newly acquired observations, the model generates selection commands and transmits them back to the robot. Subsequently, the robot executes the corresponding maneuvers using an onboard local motion planner (specifically, the off-the-shelf module provided by the Qizhi ROS platform).

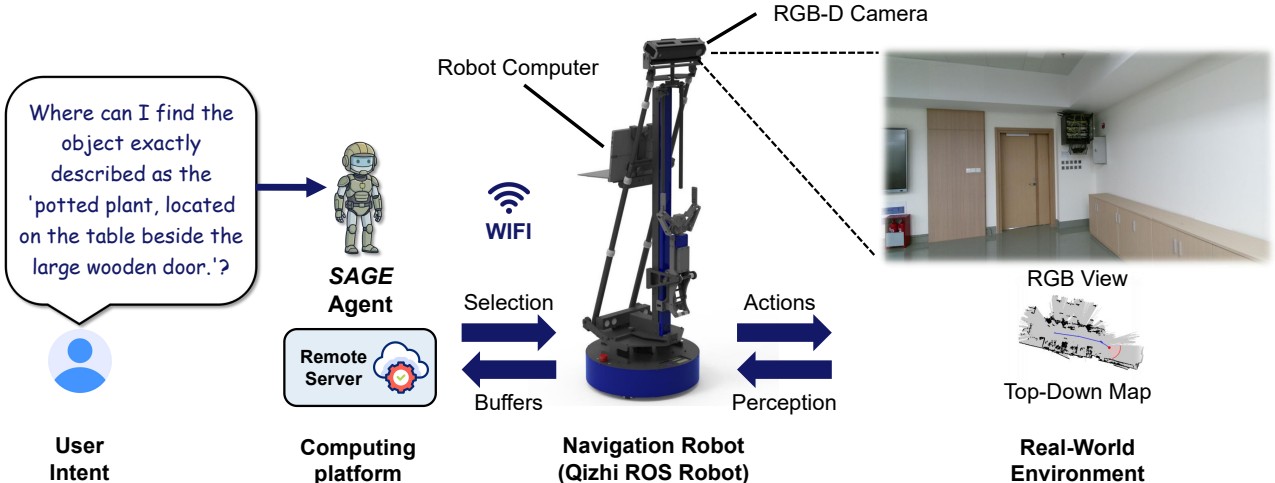

*Figure 10.* Illustration of *SAGE*'s deployment strategy for real-world robot navigation platforms.

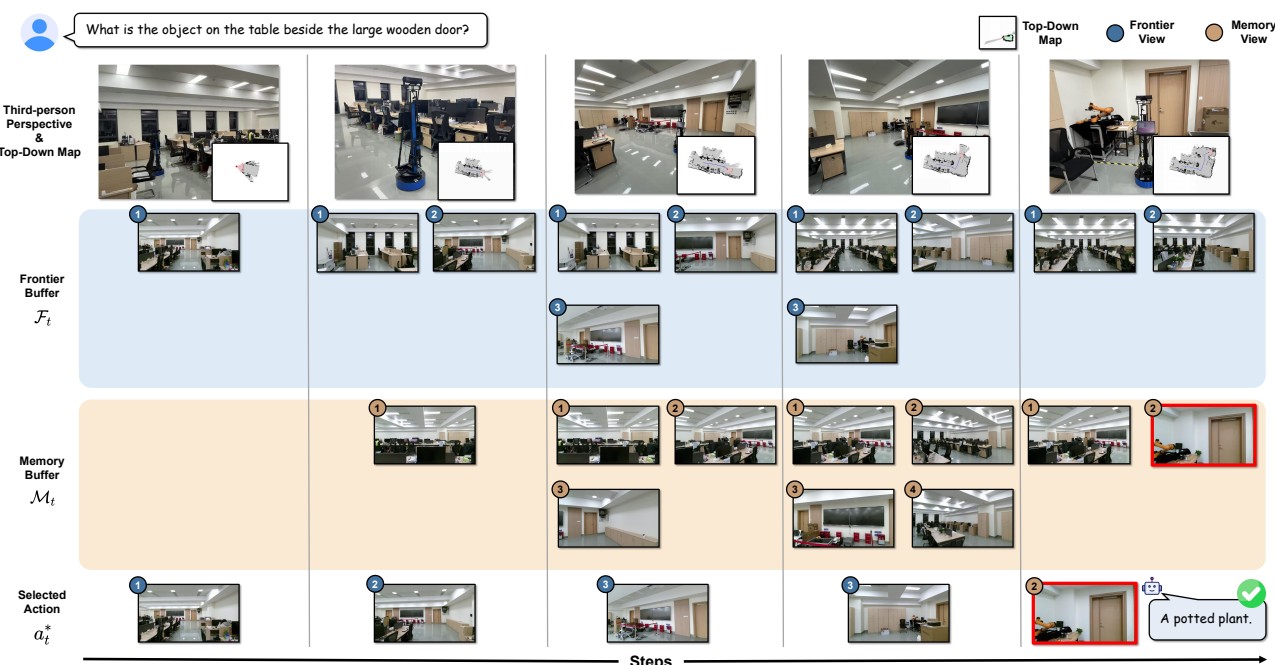

*Figure 11.* Visualization in real-world environment. Best viewed when zoomed in.

## J.2. Computational Cost and Latency Analysis

The real-world deployment relies on a client-server architecture. We provide a breakdown of the computational cost and latency for a single navigation step in Table 14. By compressing observations to $256 \times 256$ before transmission, the uplink bandwidth requirement is kept below 2 Mbps, ensuring smooth operation even on standard WiFi networks. The heavy computation is offloaded to the server; the robot's onboard computer operates at standard load, maintaining a battery life of approximately 4 hours during continuous operation.

## J.3. Visualization in the Real-World

Figure 11 visualizes a complete decision-making cycle for the query "What is the object on the table beside the large wooden door?". Despite the complex layout and visual clutter, the agent successfully selects image from the frontier buffer $\mathcal{F}_t$ and memory buffer $\mathcal{M}_t$ that effectively guides the robot toward the target. In the final step, upon recognizing the target within

its memory buffer, the agent correctly halts exploration and give correct answer.

## K. Limitations

We acknowledge several constraints of *SAGE*. (1) *SAGE* is currently optimized for indoor environments. The semantic priors derived from the sandbox, such as room connectivity and object affordances, are not directly transferable to unstructured outdoor terrains or large-scale urban navigation tasks without redefining the semantic abstraction. (2) The framework assumes quasi-static environments. The inference latency of the VLM backbone in Table 14 creates a bottleneck for real-time reaction to highly dynamic entities. This can be mitigated by optimizing VLMs with techniques such as model quantization. (3) *SAGE* is a planner-assisted high-level semantic navigation method rather than an end-to-end low-level control policy. Its performance depends on reliable frontier generation, occupancy mapping, and geometric execution. (4) Our real-world evidence is limited in scale and primarily demonstrates feasibility under indoor, quasi-static conditions.

## L. Prompt Design

We present the complete prompt for sandbox task/experience synthesis during the *Genesis* phase in Figure 12 and 13. The prompt used for experience retrieval is shown in Figure 14. Finally, Figures 15 present the prompts for training and navigation during *Evolution* and *Navigation* phases.

## Task Synthesis Prompt Template

***System Prompt:***

You are an indoor navigation task designer. Your goal is to create a visual task that is verifiable within the image based \*strictly\* on the following raw labels and scene graph data.

***Content Prompt:***

- **Detected Objects:** {objects}
- **Spatial Relationship to leverage:** {core_relationship}

Please follow these guidelines:
1. Identification: Based on the "semantic variation summary," determine the most prominent **object** or **object relationship** (e.g., a conspicuous item) that is clearly revealed only in the image.
2. Motivation: This task is important for the agent as a valuable learning point.
3. Focus: Generate a specific **natural language task** based on the discovery from Guideline 1.
4. Note that you should give a direct question and the corresponding answer that can be understood by others. Don't mention words like 'image', 'on the left of the image', etc.

Your task is to generate a Question and the corresponding Answer specifically for the Task Format:
**{target_task}**
Return your generated Question and Answer in the following format:
Task Format: [Format]
Question: [Question]
Answer: [Answer]

Example:
Task Format: functional reasoning
Question: It's too bright in the living room, how can I make it darker?
Answer: Lower the shades over the porch door.

*Figure 12.* Prompt for Sandbox Task Synthesis. The placeholders {objects} and {core_relationship} are replaced by the detected objects and the scene graph. {target_task} is replaced by the eight distinct task categories.  is replaced by the front view of the final waypoint.

## Experience Synthesis Prompt Template

### System Prompt:

You are a strategic information integration expert proficient in robotic indoor navigation analysis. Your objective is to analyze the relationship between the given Question, Current Image, Objects and Spatial Relationship, distilling it into concise navigational historical experience.

### Content Prompt:

Question: {Q}
Detected Objects: {objects}
Spatial Relationship: {core_relationship}

In order to answer the Question, the robot has selected the current image as the key trajectory step:
Current Image: 
Task: Deduce the link between this image and the potential answer.
Output Format:
IF answering "[Question]" AND observing "[Visual Cues in image]", THEN prioritize this path.
(e.g. IF answering "What is on the couch?" AND observing "a corridor leading to living room." THEN prioritize this path.)

*Figure 13.* Prompt for Sandbox Task Synthesis. The placeholders {Q}, {objects} and {core_relationship} are replaced by the synthetic task, detected objects and the scene graph.  is replaced by the front view of the final waypoint.

## Experience Template

<EXP>
Guidance from Memory:
{retrieved_experience}
(Instruction: Carefully check if any candidate image contains the visual cues mentioned in the 'IF' condition of this experience. If a match is found, strictly prioritize that path as per the 'THEN' rule.)
</EXP>

*Figure 14.* Experience template. The placeholder {retrieved_experience} is replaced by the retrieved sandbox experience.

## *Evolution & Navigation* Prompt Template

### *System Prompt:*

Task: You are an intelligent robot navigating in an indoor scene. Your task is to select a Frontier Image for further exploration or a Memory Image for answering the given Question.

### *Content Prompt:*

Question:

{Q}

{exp_section}

Definitions:

1. Frontier Image: An observation of unexplored areas that may provide new clues for answering the Question. Selecting a Frontier Image means that you will further explore that direction. If you choose a Frontier image, you need to explain why you would like to choose that direction to explore.

2. Memory Image: An observation of several known objects. Selecting a Memory Image means that you have found the final destination to answer the Question. If you choose a Memory Image, you need to directly give an answer to the question. If you don't have enough information to give an answer, then don't choose a Memory Image.

Candidate Images:

[Frontier Images]

Frontier Image 0: 

[Detected Objects]: {objects}

...

[Memory Images]

Memory Image 0: 

[Detected Objects]: {objects}

...

Instruction for Reasoning or Answer:

1. Find visual evidence for "{Q}" considering <EXP>.

2. Response constraints:

   - The Reason or Answer must be a simple, direct, natural sentence understandable by others. NO meta-words (e.g., 'memory', 'this image').

   - You should always choose the ONE most relevant Memory/Frontier index, even if you used multiple images to infer the information.

3. Provide your response in the following strict format:

   - If you choose a Frontier Image (for reasoning/exploration), return: "Frontier Image i

Reason: [Reason]"

   - If you choose a Memory Image (for answering), return: "Memory Image i

Answer: [Answer]"

   (Where i is the index of the chosen image within its category)

4. Examples:

Frontier Image 0

Reason: The hallway likely leads to the living room.

Memory Image 1

Answer: The red apple is on the white counter.

Now return your response:

*Figure 15.* Prompt for Sandbox Task Synthesis. The placeholders {Q}, {exp_section} and {objects} are replaced by the sandbox synthetic task, experience (see Figure 14) and detected objects.  are replaced by the Frontier Images and Memory Images.

