# OpenReview forum: "Plan in Sandbox, Navigate in Open Worlds: Learning Physics-Grounded Abstracted Experience for Embodied Navigation"
_ICML.cc/2026/Conference — ICML 2026 regular_

### Official Review · Reviewer_qFed · 2026-03-07

**Soundness:** 3
**Presentation:** 3
**Significance:** 2
**Originality:** 3
**Overall Recommendation:** 4
**Confidence:** 3

**Summary:**

This paper introduces Sandbox-Abstracted Grounded Experience (SAGE), which is a three-phase framework consists of Genesis phase that generates diverse tasks and uses vision-language models (VLMs) for embodied experiences, Evolution phase that internalizes embodied experiences and preserves training stability, Navigation phase that decomposes high-level plans into nodes for robust navigation behaviors. The main contribution is to use abstract and physics-grounded experience for converting high-level reasoning into low-level actions. By evaluating on two long-horizon embodied navigation benchmarks (A-EQA and GOAT-Bench), they show that SAGE achieves the state-of-the-art performance in both benchmarks. They additionally perform multiple ablation experiments to show the impact of number of input frames (i.e., 4 images in the best), different components (i.e., full components achieve the best results), and navigation phase (i.e., retrieving relevant experience is crucial for the success).

**Compliance With Llm Reviewing Policy:**

Affirmed.

**Final Justification:**

No further questions. Remain original positive score.

**Key Questions For Authors:**

No question.

**Limitations:**

yes

**Strengths And Weaknesses:**

Strengths:
1. Comprehensive experiments. Main experiments contain two long-horizon embodied navigation benchmarks (i.e., show SOTA performance over multiple baselines) and there are three ablation studies covering all aspects, and claims in the paper are well-supported (i.e., all components and designs are contributing to the success).
2. Important problem and interesting formulation. SAGE targets the bottleneck of data scarcity in embodied navigation by internalizing high-level intents into robust low-level control. Additionally, the three-phase design is especially interesting and easy to interpret the intuitions behind it. Also, the paper has good flows and illustrative figures for understanding the core designs.

Weaknesses:
1. No comparison between the efficiency of SAGE and other methods like 3D-Mem. It would be good to add such comparison to evaluate the methods in multi-dimension.
2. No qualitative example to show the failure mode of other methods in embodied navigation and how SAGE resolves these failure modes.

---

> ### Author Rebuttal · Authors · 2026-03-31
>
> Thank you for the careful reading and constructive comments. We are grateful for your recognition of our experiments and formulation. Below, we address your concerns:
>
> **W1. No comparison between the efficiency of SAGE and other methods like 3D-Mem. It would be good to add such comparison to evaluate the methods in multi-dimension.**
>
> Thank you for your helpful suggestion. We add a controlled comparison against 3D-Mem under the same Qwen3-VL-2B backbone and the same A-EQA/GOAT-Bench evaluation episodes. In this setting, SAGE uses fewer VLM calls (9.9k), lower token cost (2.5M), and shorter wall-clock time (25.0h) with the same peak GPU memory (11.0GB), while achieving substantially better SR and SPL:
> | Method | Backbone    | A-EQA SR† | A-EQA SPL† | GOAT SR | GOAT SPL | VLM Calls | Wall-Clock | Peak GPU Mem | Token Cost |
> | ------ | ----------- | --------: | ---------: | ------: | -------: | --------: | ---------: | -----------: | ---------: |
> | 3D-Mem | Qwen3-VL-2B |      44.3 |       19.4 |    46.4 |     20.3 |     11.7k |      34.8h |       11.0GB |       3.0M |
> | SAGE   | Qwen3-VL-2B |      53.2 |       37.1 |    56.7 |     38.9 |      9.9k |      25.0h |       11.0GB |       2.5M |
>
> We also clarify that Genesis/Evolution are one-time offline costs, whereas this comparison focuses on deployment-time navigation efficiency. We provide an anonymous link containing the model usage and cost across stages and methods: https://anonymous.4open.science/r/ICML_Model_Cost-F2C2 (In accordance with the ICML Anonymity and Links guidelines).
>
> **W2. No qualitative example to show the failure mode of cother methods in embodied navigation and how SAGE resolves these failure modes.**
>
> Thank you for the invaluable suggestion. We provide an anonymous link with matched qualitative comparisons against 3D-Mem on the same episodes: https://anonymous.4open.science/r/ICML-Reviewer-qFed-7356/ (In accordance with the ICML Anonymity and Links guidelines). Concretely:
>
> - **A-EQA_1, object-state query:** 3D-Mem keeps exploring after reaching the relevant room and fails to ground the queried evidence, resulting in an empty answer. SAGE instead selects the correct frontier at the critical step and grounds the answer from the relevant view.
>
> - **A-EQA_2, functional reasoning query:** 3D-Mem is distracted by an incorrect candidate view and produces a wrong answer "a red cloth". SAGE selects the correct candidate at the same decision point and grounds the functional answer correctly "using an eraser".
>
> - **GOAT-Bench_1, language-description goal:** 3D-Mem selects the wrong frontier candidate near the target area and misses the referred instance. SAGE chooses the correct candidate under the same spatial description and successfully reaches the goal.
>
> - **GOAT-Bench_2, image-goal navigation:** 3D-Mem commits to an incorrect frontier candidate and fails to localize the image-specified target. SAGE selects the matching candidate at the divergence step and successfully finds the goal.
>
> These comparisons show that SAGE's advantage over 3D-Mem is not from random favorable examples, but from more reliable critical-view selection and stronger grounding at the decision step.

---

> > ### Author Rebuttal · Reviewer_qFed · 2026-04-01
> >
> > Thanks for addressing my concerns.

---

> > > ### Author Response · Authors · 2026-04-04
> > >
> > > Thank you for your time and thoughtful feedback. We sincerely appreciate it.

---

### Official Review · Reviewer_UEkS · 2026-03-11

**Soundness:** 3
**Presentation:** 3
**Significance:** 3
**Originality:** 2
**Overall Recommendation:** 4
**Confidence:** 3

**Summary:**

This paper proposes SAGE, an experience-driven framework for embodied navigation. It attempts to bridge the visual gap between simulation and real-world environments through semantic abstraction, training agents in a physics-grounded semantic space rather than in photorealistic simulation. By generating, summarizing, and internalizing structured experience, SAGE aims to improve navigation success rates in open-world settings.

**Compliance With Llm Reviewing Policy:**

Affirmed.

**Final Justification:**

Thank you for the response. The response addresses most of my concerns.

I recommend incorporating these materials into the final version (main paper or appendix) to make the contribution more complete.

I will maintain the score for weak acceptance.

**Key Questions For Authors:**

1. The paper claims to mitigate the Sim2Real gap by replacing photorealistic discrepancies with semantic abstraction. However, the experimental section does not provide explicit Sim2Real evaluation or dedicated transfer metrics. Could the authors clarify how Sim2Real generalization is quantitatively measured in this work, and provide additional evidence demonstrating that the proposed semantic abstraction indeed improves real-world transfer compared to standard simulation-based training?

2. How efficient is data synthesis during the Genesis phase and subsequent Evolution training? In unseen scenes, is it necessary to resynthesize experiences? This is critical for the agent's navigation in unseen environments.

3. Random experience injection already yields +1.73% SR† over no retrieval (Table 4). Could you provide an experiment with deliberately mismatched experience injection to better isolate the contribution of semantic relevance? This would substantially strengthen the motivation for the vector database retrieval design.

4. Was there any exploration of annealing ε_exp during training rather than keeping it fixed? Intuitively, a tighter upper bound in later training stages might reduce instability once the policy has already absorbed sandbox priors.

5. Could you report wall-clock time for Genesis phase data synthesis (including Qwen3-VL-Plus API calls) and Evolution training on 4×A100s? This is critical for practitioners assessing whether SAGE is practical for new deployment environments.

**Limitations:**

yes

**Strengths And Weaknesses:**

Strengths:

1. The paper addresses a fundamental challenge in embodied navigation: bridging high-level VLM reasoning with low-level robot control under limited real-world interaction data.

2. The ACC mechanism is a well-motivated modification to policy optimization, enabling aggressive knowledge absorption from experience-augmented samples.

3. Extensive experiments on both A-EQA and GOAT-Bench demonstrate clear performance gains over prior SOTA methods.

Weaknesses:

1. The abstract and contribution sections claim that the paper addresses the Sim2Real gap caused by visual discrepancies between simulated and real-world environments. However, the experimental section does not provide sufficient evidence, a dedicated evaluation, or corresponding metrics to substantiate this claim.

2. The model's performance relies on IF-THEN rules based on large models, yet the paper lacks analysis and explanation regarding the reliability of such experiential acquisition methods.

3. The paper does not provide an analysis of resource efficiency across different methods, making it difficult to demonstrate that the model's performance improvement primarily stems from SAGE's mechanism rather than data scale and diversity, as shown in Figure 5b.

4. The 4B model is reported in Table 1 and Table 3 but receives almost no analysis in the text. Whether AAC and experience rule gains remain consistent at larger scale is left unexamined.

5. The full-set A-EQA evaluation (SAGE†) shows Functional Reasoning jumping from 43.2% (subset) to 56.8% (full set) while Object Recognition drops. This inconsistency across splits is not discussed, raising questions about whether the 184-question subset is representative.

6. Some design choices overlap with prior work, such as semantic abstraction of visual entities [1] and large-scale synthetic task construction [2].

[1] OVER-NAV: Elevating Iterative Vision-and-Language Navigation with Open-Vocabulary Detection and StructurEd Representation, CVPR 2024

[2] Towards Long-Horizon Vision-Language Navigation: Platform, Benchmark and Method, CVPR 2025

---

> ### Author Rebuttal · Authors · 2026-03-31
>
> We sincerely thank you for the detailed evaluation and constructive feedback. Your acknowledgment of the bridging of VLM reasoning with robot control and the well-motivated ACC mechanism is greatly appreciated. Below, we address your concerns:
>
> **W1/Q1. Sim2Real evidence is currently limited.**
>
> Our current paper does not introduce a dedicated Sim2Real benchmark or transfer metric. In our work, real-world transfer is assessed indirectly through two forms of evidence: (1) Mixed-source sandbox training consistently outperforms single-source training, which suggests lower reliance on source-specific photorealistic cues and more stable semantic and physics-grounded priors. (2) Table 4 further shows that Genesis&Evolution improves over the zero-shot VLM even without test-time retrieval, indicating that the abstracted experience is absorbed by the policy itself rather than used only as prompting.
>
> **W2. Reliability of IF-THEN rules based on large models.**
>
> We audited 200 randomly sampled IF THEN rules across HM3D and InteriorGS on both AEQA and GOAT-Bench splits:
>
>
> |Source|Sampled Rules|Correct (%)|Partially Correct (%)|Incorrect (%) |
> |-|-|-|-|-|
> |HM3D AEQA|50|42|40|18|
> |HM3D GOAT-Bench|50|38|48|14|
> |InteriorGS AEQA|50|46|44|10|
> |InteriorGS GOAT-Bench|50|62|22|16|
>
>
> Only 10-18% were incorrect or misleading, while 82-90% were at least harmless. This suggests that SAGE does not depend on every rule being exact. It benefits from aggregated experience where most retrieved rules provide usable or benign guidance, which is sufficient to support the observed gains.
>
> **W3/Q5. Resource efficiency is not reported clearly enough.**
>
> Under the same sandbox pipeline, increasing synthetic data yields only diminishing returns, while the default SAGE design still benefits from data composition (Fig. 5a) and retrieval-guided policy internalization beyond scale alone. We provide a cost table in the link:  https://anonymous.4open.science/r/ICML_Model_Cost-F2C2 (In accordance with the ICML Anonymity and Links guidelines), which reports SAGE's 3 stages cost together with the online cost of 3D-Mem.
>
> **W4. Under-analysis of the 4B model.**
>
> **At 4B, the gains remain consistent rather than disappearing.** Table 1 shows that SAGE still improves over the corresponding 4B baseline on both A-EQA and GOAT-Bench, and Table 3 preserves the same ablation ordering as 2B. Sandbox tasks provide the largest improvement, while experience rules and AAC continue to add further gains on top.
>
> **W5. Inconsistency between the subset and the full set.**
>
> **We follow the standard 3D-Mem protocol for the main A-EQA result, so the 184-question subset is used for comparability.** Table 8 shows that full-set evaluation leaves overall performance nearly unchanged (53.2/37.1; 54.1/36.2 SR/SPL) while category accuracies shift, suggesting a category-composition effect rather than a contradiction.
>
> **W6. Overlap with prior work.**
>
> OVER-NAV uses semantic abstraction for online state tracking and memory. SAGE uses it offline as transferable experience for retrieval and policy learning. LH-VLN builds benchmarks. Our sandbox builds reusable experience, not benchmarks.
>
> **Q2. Efficiency of Genesis and Evolution & Is re-synthesis needed in unseen scenes?**
>
> Please see the cost table in **W3/Q5**. **No resynthesis is required in indoor unseen scenes.** In unseen scenes, SAGE reuses the fixed experience database built in Genesis and the evolved policy to retrieve relevant priors at test time.
>
> **Q3. Mismatched experience injection**
>
> We add a mismatched-experience condition, which retrieves surface-matching candidates from the experience bank, then selects the most semantically divergent one. As shown below, mismatched injection is clearly below semantically matched retrieval, indicating that the gain is not from adding arbitrary experience text, but from retrieving task-relevant experience.
>
>
> | Method| A-EQA SR† | A-EQA SPL† | GOAT-Bench SR | GOAT-Bench SPL |
> | --- | --- | --- | --- | --- |
> | Mismatched Exp.|44.88|27.36|49.47| 29.25|
> | SAGE |53.21|37.07|56.69| 38.90|
>
> **Q4. Exploration of Annealing ε_exp During Training**
>
> We test a late-stage annealing schedule, which is 1.0 for the first 50% of training, then linearly annealed to 0.3:
> |Schedule|ε_exp|A-EQA SR†|A-EQA SPL†|GOAT-Bench SR|GOAT-Bench SPL|
> |-|-|-|-|-|-|
> |Fixed|1.0|53.21|37.07|56.69|38.90|
> |Fixed|0.3|46.23|29.47|49.60|28.52|
> |Annealed|1.0 $\rightarrow$ 0.3|51.62|38.21|55.24|37.82|
>
> It slightly improves A-EQA SPL but reduces SR and GOAT-Bench performance, while fixed 0.3 degrades substantially, suggesting that a broad early bound is important for absorbing sandbox priors and that fixed 1.0 remains the best overall choice.

---

> > ### Author Rebuttal · Reviewer_UEkS · 2026-04-02
> >
> > Thank you for the response. The response addresses most of my concerns.
> >
> > I recommend incorporating these materials into the final version (main paper or appendix) to make the contribution more complete.
> >
> > I will maintain the score for weak acceptance.

---

> > > ### Author Response · Authors · 2026-04-04
> > >
> > > Thank you for your time and thoughtful evaluation. We appreciate your constructive feedback and your recommendation.

---

### Official Review · Reviewer_eE33 · 2026-03-16

**Soundness:** 3
**Presentation:** 3
**Significance:** 3
**Originality:** 3
**Overall Recommendation:** 4
**Confidence:** 3

**Summary:**

This paper proposes SAGE, a three-stage framework for embodied navigation that aims to improve how agents explore unfamiliar environments and find task-relevant targets. The first stage, Genesis, automatically creates training experience in a simplified sandbox setting. The method generates navigation tasks together with structured guidance about what visual cues and decisions are useful for solving them. The second stage, Evolution, uses this synthetic experience to train the navigation policy so that it can make better high-level decisions during exploration. The third stage, Navigation, applies the learned policy at test time: the model decides where the agent should go next based on its current observations and memory, while a standard motion planner handles the low-level movement. Empirically, the paper reports improvements over prior RL-based and VLM-based baselines on A-EQA and GOAT-Bench. The experiments suggest that the main benefit comes from the sandbox-generated training tasks, while experience retrieval and the additional training design provide further gains. Overall, the paper's core claim is that structured synthetic experience can substantially improve high-level embodied navigation in open-world settings.

**Compliance With Llm Reviewing Policy:**

Affirmed.

**Final Justification:**

No changes to the overall score. The paper is well framed and executed. My questions were minor clarifications relating to baselines and transparency. I already had a good idea of what the authors were trying to convey. My biggest concern still remains that the low level control is essentially delegated to a planner while there are other hierarchical methods that learn end-to-end hierarchical plans without relying on external planners, reducing the scientific scope of this paper.

**Key Questions For Authors:**

1. The Navigation phase relies on planner-backed execution. Did all VLM baselines use the same frontier generation, occupancy map, and low-level shortest-path execution stack at evaluation time? If not, please clarify exactly which components are shared and which are method-specific. I want to generally understand - are the gains due to better high-level semantic choice under a common navigation stack, or are they partly due to differences in the underlying execution stack across methods?
2. Relatedly, how should readers compare SAGE to the RL baselines, given that SAGE appears to solve high-level semantic subgoal selection with planner-backed execution rather than end-to-end action generation? I would like the authors to clarify whether they view these as direct competitors or as methods operating under different assumptions.
3. In the Evolution section, the paper samples a Bernoulli mask with parameter eta_t, but later studies eta = 1.5. How exactly is eta_t converted into a valid sampling probability in implementation? If there is a clamp, normalization, or alternate interpretation, please state it explicitly.
4. The paper uses different Qwen models for different roles: Qwen3-VL-Plus for task and rule synthesis, Qwen3-VL-2B and 4B as policy backbones, and Qwen3-235B-A22B as the automatic evaluator on A-EQA. What was the rationale for these choices? I would also like a clearer accounting of compute and infrastructure for Genesis and for the judge-based evaluation, not only for Evolution training. A 235B model is rather large and I am curious how that was hosted/served.

**Limitations:**

The paper does discuss some limitations, which I appreciate. However, I think the discussion is incomplete. It should also discuss the dependence on planner-backed execution, the reliance on strong external teacher and judge models, the limited quantitative real-world evidence, and the benchmark design choices that affect evaluation fairness.

**Strengths And Weaknesses:**

## Strengths
1. I like the problem focus. The paper addresses a real bottleneck in embodied navigation - the lack of aligned data that teaches an agent which semantic cues matter for long-horizon search. The Genesis stage is the most interesting part of the paper. It creates structured supervision rather than relying only on raw simulator interaction.
2. The empirical gains are substantial. On A-EQA, SAGE with Qwen3-2B improves over 3D-Mem with the same backbone from 44.3 to 53.2 SR and from 19.4 to 37.1 SPL. The 4B variant reaches 60.2 SR and 47.2 SPL.
3. The ablations are useful. They suggest that sandbox-synthesized tasks are the main driver, retrieved experience also helps, and the experience rules plus AAC give smaller but consistent gains. I appreciate that the paper gives enough breakdown to identify where the improvements actually come from.

## Weaknesses
1. My main concern is that the learned part of the system is not low-level control. The policy selects frontier or memory nodes, and then a geometric planner executes the motion. In simulation this uses the Habitat-Sim follower, and in the real world it uses the ROS navigation stack. This makes the problem easier and changes what is being solved. I would therefore frame the contribution as high-level semantic subgoal selection under strong geometric execution, not as an advance in end-to-end embodied control. This distinction matters when comparing against RL baselines.
2. I am not fully convinced the baseline comparisons are transparent enough. The paper does not clearly state whether all VLM baselines were re-evaluated with the same frontier generation, occupancy map, and low-level shortest-path execution stack as SAGE. If not, the comparisons are less clean than the main table suggests. This is especially important because methods such as 3D-Mem already live in a related planner-assisted regime, while the RL baselines seem to belong to a different agent class altogether.
3. There is a technical inconsistency in the Evolution description. The paper says it samples a Bernoulli mask with parameter eta_t, but later studies fixed high guidance at eta = 1.5. A Bernoulli probability above 1 does not make sense as written. This may be a notation issue, but it needs clarification because it directly affects reproducibility.
4. The strongest parts of the pipeline rely on stronger external models than the headline training setup suggests. The policy backbones are Qwen3-VL-2B and 4B, but task and rule synthesis uses Qwen3-VL-Plus, and A-EQA evaluation uses Qwen3-235B-A22B as the judge. I do not object to this, but the rationale and cost are not discussed clearly enough.
5. The paper sometimes overstates its conceptual novelty. I do think the sandbox-generated structured supervision is useful. However, the work is still closer to modular planner-assisted embodied navigation than to a new class of hierarchical RL or predictive world-model learner. In particular, the policy does not learn temporally extended options in the sense of FeUdal Networks or Option-Critic, and it does not learn a predictive generative simulator of future observations in the sense of Navigation World Models.

---

> ### Author Rebuttal · Authors · 2026-03-31
>
> Thank you for your careful reading of our manuscript and for the constructive comments. We sincerely appreciate your recognition of the value of the Genesis stage for structured supervision. Below, we address your concerns:
>
> **W1. Task framing and comparison fairness.**
>
> SAGE is a modular high-level semantic navigation method rather than an end-to-end low-level policy. In Navigation, the learned module selects a frontier or memory target, while geometric execution is handled by the geometric planner. In our revision, we will reframe the paper's core contribution as high-level semantic subgoal selection under strong geometric execution.
>
> **W2. Baseline transparency and stack comparability.**
>
> We appreciate the reviewer's concern. To clarify, all VLM baselines were re-evaluated using the same frontier generation, occupancy map construction, and low-level shortest-path execution stack as SAGE, so the VLM comparisons in the main table are controlled at the execution layer. In particular, 3D-Mem is our closest comparator. By contrast, the RL baselines belong to a different agent class and are included as cross-paradigm references rather than identical-stack comparisons.
>
> **W3/Q3. Clarification of $\eta_t$ and the valid sampling probability.**
>
> Thank you for carefully reviewing our manuscript and pointing out the issues. The reported value $1.5$ actually refers to the target validation score $R_{\text{target}}$ used in the dynamic schedule for updating $\eta_t$, rather than to the Bernoulli parameter itself. To obtain a valid sampling probability, we compute:
>
> $
> \eta _ t=\max\left(\eta _ {\min}, \eta _ {\mathrm{init}}\left(1-\frac{\min(R _ {\mathrm{val}}^{(t)}, R _ {\mathrm{target}})}{R _ {\mathrm{target}}}\right)\right),
> $
>
> followed by $m_t \sim \mathrm{Bernoulli}(\eta_t)$. In our experiments, $\eta_{\text{init}} = 0.8$, $\eta_{\min} = 0.0$, and $R_{\text{target}} = 1.5$. No extra clamp or normalization is used. We will address this in the revised version.
>
> **W4/Q4. Model-role rationale and cost transparency.**
>
> Qwen3-VL-2B and 4B are the actual policy backbones in Evolution and Navigation, which determine deployment latency. Qwen3-VL-Plus is used only offline in Genesis for task and rule synthesis, where higher semantic quality is preferred and the cost is amortized over the generated training set. Qwen3-235B-A22B is used only once as an offline A-EQA judge, not for training or deployment. In our setup, Genesis and the 235B judge are served through remote Alibaba Cloud APIs, while Evolution training and online Navigation use a local OpenAI-compatible endpoint. On average, Genesis uses about 108k VLM calls and 37.5M tokens in total, online Navigation about 9.9k calls and 2.5M tokens, and A-EQA judging 138 calls and 137.6K tokens. **Due to the space limitations of the rebuttal, we provide an anonymous link containing the model usage and cost across stages:** https://anonymous.4open.science/r/ICML_Model_Cost-F2C2 (In accordance with the ICML Anonymity and Links guidelines).
>
> **W5. Scope of conceptual novelty.**
>
> Thanks for your invaluable comment. SAGE is viewed as a modular embodied navigation framework, not as a new class of hierarchical RL or a predictive world-model learner. The learned component does not learn temporally extended options in the FeUdal or Option-Critic sense, and it does not learn a generative simulator of future observations. **The novelty lies in showing that structured supervision from the sandbox pipeline can substantially improve high-level semantic decision-making in embodied navigation.**
>
> **Q1.  Clarification on shared and method-specific execution components**
>
> **All VLM baselines were re-evaluated using the same frontier generation, occupancy map updates, and low-level shortest-path execution stack as SAGE.** The method-specific components are the visual representation, memory or retrieval design, prompting, and the high-level policy for selecting candidate targets. Therefore, within the VLM group, the gains should be attributed primarily to better high-level semantic target selection under a common navigation stack. RL baselines belong to a different agent class and are included only as cross-paradigm references.
>
> **Q2. How should readers compare SAGE to the RL baselines?**
>
> SAGE and the RL baselines should not be viewed as direct competitors at the control level. SAGE solves high-level semantic target selection, while the RL baselines generates end-to-end actions. **We include them as cross-paradigm references on the same embodied navigation tasks and end metrics.** The intended claim is that stronger high-level semantic choice can substantially improve end-task performance in a planner-assisted setting, not that SAGE is superior in low-level control.
>
> **Limitation. The current discussion omits several scope conditions.**
>
> Thank you for your suggestion. Due to space constraints in the ICML rebuttal, we will provide further discussions in the revised version.

---

> > ### Author Rebuttal · Reviewer_eE33 · 2026-04-03
> >
> > The rebuttal adequately addresses my main clarification questions. I appreciate the authors' clearer reframing of SAGE as a modular high-level semantic navigation method rather than an end-to-end control method. All VLM baselines being re-evaluated with the same frontier generation, occupancy map, and low-level execution stack resolves my main fairness concern within the VLM group. Overall, the rebuttal improves the paper's calibration and clarity, but it does not materially change my assessment, so I will keep my original scores and recommendation unchanged.

---

> > > ### Author Response · Authors · 2026-04-04
> > >
> > > Thank you for the careful reading and thoughtful feedback. We appreciate your time and consideration.

---

### Decision · Program_Chairs · 2026-04-30

**Decision:**

Accept (regular)

**Comment:**

The submission proposes an experiential framework for embodied navigation. It links the gap between simulation and real-world environments via semantic abstraction, with training the agent in a physics-grounded semantic space, followed by deployment.

All reviewers support the submission. The reviewers were specifically asked to comment 1) if the paper is technically sound and 2) why they'd find it exciting. The rebuttal was also incorporated. No fundamental flaw was uncovered in the review process. While the reviewers are supportive and find several aspects of the paper, e.g., planning in sandbox and Genesis interesting, the overall conclusion was that the paper can benefit from more work if it were to be transformational and lead to a change in common practices. This is something for the authors to consider for the camera-ready. Please carefully take into account all reviewers' comments and exchanges. Nevertheless, the submission has enough merit to warrant appearing in ICML.